# Agonistic Bivalent Human scFvs-Fcγ Fusion Antibodies to OX40 Ectodomain Enhance T Cell Activities against Cancer

**DOI:** 10.3390/vaccines11121826

**Published:** 2023-12-07

**Authors:** Kodchakorn Mahasongkram, Kantaphon Glab-ampai, Kanasap Kaewchim, Thanatsaran Saenlom, Monrat Chulanetra, Nitat Sookrung, Oytip Nathalang, Wanpen Chaicumpa

**Affiliations:** 1Center of Research Excellence in Therapeutic Proteins and Antibody Engineering, Department of Parasitology, Faculty of Medicine Siriraj Hospital, Mahidol University, Bangkok 10700, Thailand; kodchakorn.mah@mahidol.ac.th (K.M.); kantaphon.gla@mahidol.edu (K.G.-a.); kanasap.kaw@alumni.mahidol.ac.th (K.K.); thanatsaran.sae@mahidol.ac.th (T.S.); monrat.chl@mahidol.edu (M.C.); nitat.soo@mahidol.ac.th (N.S.); 2Graduate Program in Immunology, Department of Immunology, Faculty of Medicine Siriraj Hospital, Mahidol University, Bangkok 10700, Thailand; 3Biomedical Research Incubator Unit, Department of Research, Faculty of Medicine Siriraj Hospital, Mahidol University, Bangkok 10700, Thailand; 4Graduate Program in Biomedical Sciences, Faculty of Allied Health Sciences, Thammasat University, Rangsit Campus, Pathum Thani 12120, Thailand; oytipntl@hotmail.com

**Keywords:** cancer immunotherapy, bivalent scFvs-Fcγ antibodies, fusion protein, human single-chain antibody, OX40, OX40 agonistic antibodies, phage-display technology, ovarian cancer

## Abstract

(1) Background: Understanding how advanced cancers evade host innate and adaptive immune opponents has led to cancer immunotherapy. Among several immunotherapeutic strategies, the reversal of immunosuppression mediated by regulatory T cells in the tumor microenvironment (TME) using blockers of immune-checkpoint signaling in effector T cells is the most successful treatment measure. Furthermore, agonists of T cell costimulatory molecules (CD40, 4-1BB, OX40) play an additional anti-cancer role to that of checkpoint blocking in combined therapy and serve also as adjuvant/neoadjuvant/induction therapy to conventional cancer treatments, such as tumor resection and radio- and chemo- therapies. (2) Methods and Results: In this study, novel agonistic antibodies to the OX40/CD134 ectodomain (EcOX40), i.e., fully human bivalent single-chain variable fragments (HuscFvs) linked to IgG Fc (bivalent HuscFv-Fcγ fusion antibodies) were generated by using phage-display technology and genetic engineering. The HuscFvs in the fusion antibodies bound to the cysteine-rich domain-2 of the EcOX40, which is known to be involved in OX40-OX40L signaling for NF-κB activation in T cells. The fusion antibodies caused proliferation, and increased the survival and cytokine production of CD3-CD28-activated human T cells. They showed enhancement trends for other effector T cell activities like granzyme B production and lysis of ovarian cancer cells when added to the activated T cells. (3) Conclusions: The novel OX40 agonistic fusion antibodies should be further tested step-by-step toward their safe use as an adjunctive non-immunogenic cancer immunotherapeutic agent.

## 1. Introduction

Immunotherapy (the use of immunological factors or an immune stimulator/modifier to treat or prevent diseases) is one of the most promising approaches for the treatment of diseases including inflammations and cancers. For cancers, the immunotherapeutic agents harness or boost the immune system to work more efficiently, making it more specific and/or more able to recognize and attack the targets [1]. Immunotherapy can stand alone or be performed alongside, after, or prior to conventional cancer treatments (surgery, radiotherapy, and chemotherapy), i.e., it can serve as an adjuvant, neoadjuvant, or induction therapy [2,3,4]. In many instances, immunotherapy can work for cancers when other treatment measures do not or cannot work, such as in unresectable or radio- and/or chemo-resistant cancers [5]. 

Nowadays, there are several types of immunotherapies that have been practiced that suit different cancers including the use of adoptive immune cells, i.e., tumor-infiltrating lymphocytes [6], engineered CAR-T/Natural Killer (NK) cells [7,8], and T cell receptor (TCR)-engineered T cells [9]; cancer vaccines (either for the treatment of existing cancer or to prevent cancer development) [10,11]; immunomodulators, i.e., cytokines, Bacillus Calmette–Guérin (BCG), and biological response modifiers (e.g., Imiquimod, Lenalidomide) [12,13,14,15]; immune system agonists [16,17]; and monoclonal antibodies (mAbs) [18,19]. Usually, the immune modifiers, immune system agonists, and monoclonal antibodies do not kill cancer cells directly but work by enhancing and/or helping the immune system to better find cancers in the body (marking cancerous cells) for the immune system to attack them.

Tumorigenesis involves cancer immunoediting, which is a continual process consisting of three phases [20,21,22]. The first phase involves the initially transformed body cell that, if not eliminated by the host immune surveillance of natural killer (NK) cells, can manage to form a nascent tumor; this will be followed by the second phase, a relatively prolonged equilibrium phase in which the tumor outgrowth is immunologically restrained but not eliminated; and, finally, the third phase involves immunological sculpting of the tumor and establishment of a tumor microenvironment (TME) that favors tumor metabolism and growth, leading to tumor progression/metastasis and clinical manifestations [22,23,24]. The TME is a complex milieu consisting of vasculature (blood and lymph vessels), fibroblasts, mesenchymal stem cells (MSCs), adipocytes, neuroendocrine (NE) cells, immune cells (myeloid cells, lymphocytes), extracellular matrix, and vesicles as well as physical and chemical factors that contribute to low extracellular pH (acidosis), hypoxia, and elevated interstitial fluid pressure [25]. The immune cells in the TME play different roles at different stages of tumor development, i.e., tumor suppression in the early stage by some cell types, but tumor growth promotion at the later stage by others [20,21,22,25]. In the early stage of cancer, NK cells, CD8^+^ cytotoxic T lymphocytes (CTL), Th1 cells, and antigen-presenting cells (APCs), like dendritic cells (DCs) and M1 macrophages, are effective in suppressing tumor growth. Cytotoxic T cells and NK cells induce apoptosis, necrosis, and growth arrest of the cancerous cells by releasing IFN-γ, perforin, granzymes, etc. The apoptotic cancer cells and released components are phagocytosed/endocytosed by APCs, which process and present the antigenic peptides to lymphocytes in the lymphoid tissues for adaptive immune responses against the tumor. However, in the later phase, tumors can evade the immune defenses and progress by creating an immunosuppressive TME containing M2 tumor-associated macrophages (TAMs), CD4^+^ Th2 cells, and, especially, regulatory T cells (Tregs) [25]. Tregs in the TME attenuate/inhibit effector T cells/NK cells and DCs by producing a myriad of immunosuppressive factors (both soluble and cell-bound) including immunosuppressive cytokines (like IL-10, TGF-β, and IL-35); high levels of cell-surface IL-2R (CD25) that deprive the microenvironment of IL-2, which is important for the proliferation and survival of effector T cells, esp. CD8^+^; surface-exposed ectonucleotidases (CD39 and CD73) that generate adenosine for impairment of effector cell functions; and, most of all, surface expressed immune-checkpoint molecules (e.g., PD1, CTLA-4, and LAG-3) that bind to respective receptors/ligands (PD-L1, CD86, and MHC-II, respectively) on the APCs, causing the refractoriness of the effector cells [26]. The blockage of immune-checkpoint signaling pathways in the effector T cells has shown promising results in cancer immunotherapy [27,28,29]. Monoclonal antibodies targeting immune checkpoints, e.g., Nivolumab for PD-1, Avelumab for PD-L1, Ipilimumab for CTLA-4, Relatlimab for LAG-3, or combined treatment with these have been approved for the immunotherapy of many cancers [30].

Refractory effector T cells in the TME can be revitalized to regain their anti-tumor activities, i.e., the ability to restrain tumor growth, by stimulating T cell costimulatory receptors, such as those belonging to the tumor necrosis factor receptor superfamily (TNFRSF), i.e., CD40, 4-1BB, and OX40 [31,32]. Ligation of OX40 (TNFRSF4, CD134) on the effector T cell surface by OX40 agonists causes the receptor to oligomerize (cluster) and stimulates intracellular signaling, which enhances anti-tumor immunity leading to therapeutic effects in cancers [33,34,35]. Resting T cells do not constitutively express OX40 on the cell surface but transiently upregulate it upon being activated [31]. The interaction between OX40 on T cells and OX40L (CD252) expressed on antigen-presenting cells, such as DCs, macrophages, and B cells, enhances T cell expansion, trafficking, cytokine release and survival, and the generation of long-lived memory T cells [31,32]. Monoclonal antibodies that trigger OX40 signaling have been developed and tested in preclinical trials and in patients with advanced solid tumors; these were found to be well-tolerated and effective [36,37]. The treatment of tumor-bearing hosts with OX40 agonistic antibodies (using a variety of antibody formats) resulted in tumor regression [38,39,40]. The mechanisms of those OX40 agonistic antibodies against tumors involved the promotion of antigen-specific effector T cell expansion and survival in the TME and macrophage activation, as well as the reverse immune suppression of Tregs [38,41]. Regulatory T cells constitutively express OX40. OX40 signaling controls Treg proliferation and their suppressive activities [42,43]. OX40 agonists alone or in combination with immune-checkpoint blockers have high therapeutic potential for patients with advanced malignancies. Despite numerous preclinical and clinical trials of several OX40 agonists, none of these have reached clinical use [44]. The lack of sufficient potency of the OX40 antibodies in a therapeutic setting may be due to their inefficiency or low potency in properly clustering the OX40 molecules on the T cell surface in vivo [44]. The OX40 agonist-OX40 complexes must be able to cause clustering of the spatially distant OX40 molecules on the cell surface to enable their intracytoplasmic OX40 tails to generate a hexameric TRAF complex for downstream signaling and activation of the NK-κB and NFAT transcription factors, and this requires the optimal positioning of the complex on the cell surface [44]. The lack of this intrinsic property renders the OX40 agonists ineffective in therapy [44]. In this study, engineered human OX40 agonistic antibodies that target the human OX40 ectodomain in the form of bivalent human single-chain variable fragments (HuscFvs) linked to the human IgG1-Fc portion (fusion antibodies) were generated. The production, characterization, and evaluation of these novel OX40 agonistic antibodies in T cell stimulation to enhance anti-tumor activities form the core of this article.

## 2. Materials and Methods

### 2.1. Ethics Statement 

Experiments involving human samples were approved by the Institutional Review Board of the Faculty of Medicine Siriraj Hospital, Mahidol University, Bangkok, Thailand (IRB Si 651/2018). 

### 2.2. Cells and Culture Media

Human embryonic kidney (HEK) 293T cells and Jurkat T cells (a leukemic cell line isolated from peripheral blood of a patient with acute T lymphocytic leukemia) were from the American Type Culture Collection (ATCC; Manassas, VA, USA). HEK293E suspension cells were from IBA Lifesciences (Göttingen, Germany). The ovarian cancer cell line, SK-OV-3, was from Dr. Somponnat Sampattavanich, Department of Pharmacology, Faculty of Medicine Siriraj Hospital, Mahidol University, Bangkok. Human peripheral blood mononuclear cells (PBMCs) were isolated from blood of healthy donors. HEK293T and SK-OV-3 cells were cultured in Dulbecco’s modified Eagle’s medium (DMEM) (Gibco, Thermo Fisher Scientific, Waltham, MA, USA) supplemented with 10% fetal bovine serum (FBS) (HyClone; Cytiva, Marlborough, MA, USA), 100 units/mL penicillin, 100 mg/mL streptomycin, and 2 mM L-glutamine (Gibco, Thermo Fisher Scientific). HEK293E suspension cells were cultured in CDM4HEK293 serum-free medium (Cytiva, Marlborough, MA, USA) supplemented with 100 units/mL penicillin, 100 mg/mL streptomycin, and 4 mM L-glutamine (Gibco, Thermo Fisher Scientific). Jurkat cells were cultured in RPMI-1640 (Gibco, Thermo Fisher Scientific) supplemented with 10% FBS (HyClone), 100 units/mL penicillin, 100 mg/mL streptomycin, and 2 mM L-glutamine (Gibco, Thermo Fisher Scientific) (complete RPMI-1640). T cells, monocytes, and dendritic cells (DCs) were cultured in AIM-V (Gibco, Thermo Fisher Scientific) supplemented with 10% FBS (HyClone), 100 units/mL penicillin, 100 mg/mL streptomycin, and 2 mM L-glutamine (Gibco, Thermo Fisher Scientific) (complete AIM-V). Human PBMCs were cultured in either complete RPMI-1640 or complete AIM-V.

### 2.3. Recombinant Ectodomain of Human OX40 

A recombinant human OX40 ectodomain (EcOX40) with inherent OX40 ligand binding activity (Abcam, Cambridge, UK) was used as the antigenic bait in phage panning to fish out HuscFv-displaying phage clones from a previously constructed HuscFv phage display library [45] for the production of HuscFvs to human EcOX40. 

A recombinant human glycosylated EcOX40 produced by transformed HEK293E suspension cells in our laboratory was used as an antigen for checking the binding of antibodies to human EcOX40. An amplicon of the EcOX40 coding sequence was ligated to a mammalian expression vector (pDSG, IBA Lifesciences) and the recombinant vector was transformed into HEK293E suspension cells. The cells were grown and the soluble EcOX40 protein was harvested from the cell culture supernatant.

### 2.4. Preparation of Mammalian Cells That Overexpress Human OX40 (HEK-OX40 Cells)

Peripheral blood mononuclear cells (PBMCs) were isolated from 10 mL of heparinized whole blood from a healthy subject by gradient centrifugation using Ficoll Hypaque (Histopaque-1077, Sigma-Aldrich, St. Louis, MO, USA). The isolated PBMCs (1 × 10^6^ cells) and Jurkat cells were cultured in complete RPMI-1640 medium (Gibco, Thermo Fisher Scientific) containing mitogens, i.e., 0.5 μg/mL of phorbol 12-myristate 13-acetate (PMA, Sigma-Aldrich, USA) and 1 μM of ionomycin (Iono, Sigma-Aldrich, USA) or 1 μg/mL phytohemagglutinin-L (PHA-L, Biochrom, Cambridge, UK) at 37 °C in a 5% CO_2_ atmosphere for 3 days. The activated cells and respective resting cells (no mitogen activated counterparts) were then stained with anti-OX40-Alexa Fluor 647 to check for OX40 expression by flow cytometry. The activated PBMCs were found to express OX40 more than the Jurkat cells; therefore, the PBMCs were used as a source of the OX40 coding gene (*tnfrsf4*). 

Human PBMCs (1 × 10^6^ cells) were cultured in complete RPMI-1640 medium (Gibco, Thermo Fisher Scientific) containing 0.5 μg/mL of PMA (Sigma-Aldrich) and 1 μM of Iono (Sigma-Aldrich) or 1 μg/mL PHA-L (Biochrom) at 37 °C in a 5% CO_2_ atmosphere for 6 h. Total RNA was extracted from the mitogen-activated cells by using Trizol reagent (Invitrogen, Waltham, MA, USA) and reversely transcribed into complementary DNA (cDNA) by using the RevertAid First Strand cDNA Synthesis kit (Thermo Fisher Scientific). The cDNA was used as the template in a polymerase chain reaction (PCR) for the amplification of the full-length human OX40 gene (human *tnfrsf4*) by using specific primers with 5′ *Xho*I and *Eco*RI restriction sites, i.e., forward primer: 5′-TACTATCTCGAGGCCACCATGTGCGTGGGGGCTCG-3′ and reverse primer: 5′-TATCATGAATTCGCGATCTTGGCCAGGGTGGAGTG-3′, from the GenBank database (NM_003327). The nucleotide sequence of the amplified PCR product was verified using Sanger sequencing (1st Base DNA sequencing service, Selangor, Malaysia) before ligation into the pJET1.2 cloning vector (Thermo Fisher Scientific) using T4 ligase (New England BioLabs, Ipswich, MA, USA). The recombinant vector was transformed into JM109 *Escherichia coli*. The preparation was spread onto Luria-Bertani (LB) agar containing 100 µg/mL ampicillin (LB-A agar plate) and incubated at 37 °C overnight. The plasmid-transformed *E. coli* colonies (clones) that grew on the agar were screened for the target gene, i.e., full-length human *tnfrsf4* by direct colony PCR using forward and reverse pJET1.2 primers, and the amplicon was sequenced (1st Base). 

For preparing mammalian cells that overexpressed full-length human OX40, bi-cistronic expression vector that expressed full-length human OX40 and the *Zoanthus* green fluorescent protein (ZsGreen1) was prepared. A commercially synthesized *MCS-P2A* nucleotide sequence and the pLVX-puro vector (Clontech, Mountain View, CA, USA) were similarly digested with *Xho*I and *Xba*I restriction endonucleases (Thermo Fisher Scientific). The cut *MCS-P2A* was ligated with the cut pLVX-puro vector using T4 ligase. The recombinant vector was transformed into DH5α *E. coli.* The transformed bacteria were spread on the selective LB-A agar plate and incubated at 37 °C overnight. The *E. coli* clones carrying the recombinant *MCS-P2A*-pLVX-puro plasmid vector were screened by direct colony PCR using forward and reverse MSCV primers. The recombinant plasmids were extracted from the DH5α *E. coli* and verified by DNA sequencing (1st Base).

A DNA fragment coding for the *Zoanthus* green fluorescent protein (*ZsGreen1*) (Genscript, Piscataway, NJ, USA) and the *MCS-P2A*-pLVX-puro vector were similarly digested with *Not*I and *Xba*I restriction enzymes. The cut *ZsGreen1* fragment was ligated with the cut *MCS-P2A*-pLVX-puro vector using T4 ligase and transformed into DH5α *E. coli.* The transformed bacteria were spread onto the LB-A agar plate. After incubation at 37 °C overnight, bacterial colonies were screened by direct colony PCR as above for the clones that harbored the *MCS-P2A-ZsGreen1*-pLVX-puro vector using the forward and reverse CMV primers. The verified *MCS-P2A-ZsGreen1-*pLVX*-*puro vector and the *tnfrsf4-*pJET1.2 were cut similarly with *Xho*I and *Eco*RI enzymes (Thermo Fisher Scientific), ligated using T4 ligase, and transformed into DH5α *E. coli*. The *E. coli* clones carrying the *tnfrsf4-P2A-ZsGreen1*-pLVX-puro vector were screened by direct colony PCR using the forward and reverse MSCV primers. The *tnfrsf4*-*P2A-ZsGreen1*-pLVX-puro vector was extracted from the *E. coli* clone by using the EndoFree Maxi Plasmid kit (Tiangen Biotech, Beijing, China). 

Human embryonic kidney (HEK293T) cells were cultured in 10% tetracycline-free-FBS-supplemented DMEM before their use in lentivirus preparation. The HEK293T cells were co-transfected with the *tnfrsf4*-*P2A-ZsGreen1*-pLVX-puro vector and Lenti-X packaging single shots (VSV-G) (Clontech), and the cells were incubated at 37 °C in a 5% CO_2_ incubator for 3 days. The culture supernatant was tested for the presence of lentivirus particles by adding Lenti-X GoStix (Clontech). A positive Lenti-X GoStix result implied that there were more than 5 × 10^5^ infectious units (IFU)/mL of lentivirus particles in the culture supernatant. The lentivirus particles in the culture supernatant were concentrated by using Lenti-X concentrator (Clontech) and checked for quantity and infectivity. For this experiment, the HEK293T cells (3 × 10^4^ cells) were added to the wells of a 96-well-tissue culture plate (Corning, Steuben County, NY, USA) and the plate was kept at 37 °C in a 5% CO_2_ incubator overnight. A five-fold serially diluted lentivirus preparation (50 μL) was added to individual wells containing the HEK293T cells and incubated further for 4 h before the addition of 10% FBS supplemented-DMEM (150 μL/well) and was then kept in the incubator for 3 more days. The cells from all wells were collected separately and the expression of the ZsGreen1 protein was determined (excitation/emission at 493/505 nm) via flow cytometry. The percentage of ZsGreen1-positive cells was used for calculating the number of lentivirus infectious units or total transducing units (TU)/mL.

HEK293T cells (1 × 10^4^ cells/well of a 96-well tissue culture plate maintained at 37 °C in a 5% CO_2_ incubator overnight) were added together with the lentivirus (MOI 10; by mixing the lentivirus with 10% FBS-supplemented DMEM in a total volume of 50 μL). The infected HEK293T cells were incubated for 4 h, followed by the addition of 10% FBS-supplemented DMEM (150 μL/well) and the cells were incubated further for 3 more days. The cells were observed for ZsGreen1 protein by using an inverted fluorescent microscope. A positive result indicated that the cells were infected with the lentivirus and that the transfected DNA was integrated into the chromosomes. Thereafter, 2 μg/mL puromycin (Invivogen, San Diego, CA, USA) was added to the cultured cells. The spent medium was replaced with fresh medium every 2–3 days. On days 7–10, non-infected cells were dead. The viable cells were cultured further in 10% FBS-supplemented DMEM. The expression of human OX40 on the cell surface was determined. The cells were stained with an Alexa Fluor 647-conjugated anti-human OX40/CD134 antibody (clone Ber-act35, Biolegend, San Diego, CA, USA) and analyzed via flow cytometry (BD LSRFortessa, BD biosciences, San Jose, CA, USA) in comparison with non-infected HEK293T cells (negative control). The HEK293T cells expressing human OX40 were designated “HEK-OX40 cells”.

### 2.5. Production of HuscFvs That Bound to Human EcOX40 by Using Phage Display Technology 

The HuscFv phage display library used in this study was constructed previously [45,46]. In this phage library, complete phage particles display HuscFvs as fusion partners of the phage coat P3 protein and contain the respective HuscFv genes (*huscfvs*) in their genomes. The diversity of the HuscFvs of this library was ~2.6 × 10^8^ [45]. After one cycle of the library propagation in TG1 *E. coli*, ~2.6 × 10^12^ cfu/mL of complete phage particles were obtained [45].

For the selection of HuscFvs displaying phage clones that bound to human EcOX40, recombinant human EcOX40 (Abcam) and HEK-OX40 cells were used as the antigenic baits in the phage panning to fish out the EcOX40-bound HuscFv-displaying phages from the HuscFv phage display library. For panning with the recombinant human EcOX40, the recombinant human EcOX40 and unrelated proteins, including bovine serum albumin (BSA, Calbiochem, San Diego, CA, USA), and skim milk (HiMedia, Maharashtra, India) were applied to separate wells in an EIA/RIA microplate (Corning) (2 μg protein in 100 μL of phosphate buffered saline, pH 7.4 (PBS)) and the plate was kept at 4 °C overnight. After discarding the fluid in all the wells, the coated wells were washed three times with PBS containing 0.05% polysorbate (Tween)-20 (PBST; 200 μL PBST/well each time). Then, each well was blocked with 200 μL of 3% BSA in PBS and the plate was incubated at 37 °C for 1 h. After discarding the excess BSA and washing the wells with PBST, the HuscFv phage display library (50 μL) was added to the skim milk-coated well and incubated at 37 °C for 1 h. The supernatant was then moved to the BSA-coated well and incubated for 1 h before the fluid containing skim milk and the BSA-subtracted phage library were transferred to the human EcOX40-coated well and kept at 37 °C for 1 h. The fluid was discarded and the well was washed thoroughly with PBST. Log-phase-grown HB2151 *E. coli* (200 μL) was added to the well containing the EcOX40-bound phages and kept at 37 °C for 15 min to allow the phages to infect the *E. coli.* The phage transformed preparation was diluted appropriately with LB broth and spread onto 2× YT-ampicillin agar (2YT-A) plates containing 2% glucose (2YT-AG agar) and the plates were incubated at 37 °C overnight.

For the selection of HuscFv-displaying phage clones that bound to human EcOX40 expressed on the surface of the HEK-OX40 cells, a cell-based phage panning was performed. The skim milk and BSA-subtracted HuscFv phage display library was prepared as above. The subtracted library (100 μL) was mixed with 1 × 10^6^ HEK293T cells in a microcentrifuge tube and kept at 4 °C on a see-saw rocker for 1 h. The cells were sedimented by centrifugation at 10,000× *g* for 30 s. The supernatant containing the HEK293T cell-unbound phage was transferred to another centrifuge tube containing 1 × 10^6^ HEK-OX40 cells. After rocking them at 4 °C for 1 h, the cells were collected by centrifugation as above and washed ten times with 1% BSA in PBS containing 0.02% NaN_3_, followed by washing with cold PBS five times. Log-phase-grown HB2151 *E. coli* (200 μL) were then added to the washed cells and the preparation was kept at 37 °C for 15 min. The phage-transformed HB2151 *E. coli* preparation was diluted with LB broth, plated onto the 2× YT-AG agar plates, and the plates were incubated at 37 °C overnight. 

The phage-infected HB2151 *E. coli* clones from the phage panning with recombinant human EcOX40 and HEK-OX40 cells that grew on the 2× YT-AG agar plates were screened by direct colony PCR for the genes coding for HuscFvs (*huscfvs*) by using R1 and R2 primers specific for the pCANTAB5E phagemids [45]. The *huscfv*-positive *E. coli* clones were grown individually in 2× YT-A broth at 37 °C with shaking aeration (250 rpm) until the optical density (OD) at 600 nm reached about 0.3–0.5. The bacterial cultures were centrifuged (4000× *g*, 4 °C, 15 min), the supernatants were discarded, and the bacteria in each pellet were resuspended in fresh 2× YT-A broth containing 0.5 mM isopropyl β-d-1-thiogalactopyranoside (IPTG) and incubated further at 30 °C, 250 rpm for 4 h. The bacterial cells of each culture were collected by centrifugation, resuspended in 1 mL PBS, and homogenized by sonication. Bacterial homogenates were centrifuged (12,000× *g*, 4 °C, 15 min). Then, the bacterial lysates (supernatants) were collected and the presence of E-tagged-HuscFvs in the lysates was determined by Western blot analysis using an anti-E-tag antibody (1:5000, Abcam, Cambridge, UK) to probe the HuscFvs in the SDS-PAGE-separated bacterial lysates.

### 2.6. Characterization of the HuscFvs

Binding of the HuscFvs to human EcOX40 was tested using an indirect ELISA and flow cytometry. For the indirect ELISA, the recombinant human EcOX40 (Abcam) and BSA (control antigen) were dissolved in PBS (5 μg/mL), and 100 μL aliquots were added to separate wells of an EIA/RIA plate (Corning). The plate was kept at 4 °C overnight. Then, all wells were washed three times with PBS before blocking each well with 200 μL of 5% skim milk in PBS at 37 °C for 1 h. The blocking solution was removed, the wells were washed with PBS, and 100 μL of the *E. coli* lysate containing HuscFvs was added to the wells coated with EcOX40 and BSA. The original HB2151 *E. coli* lysate (HB) was included in the assay as the background binding control (no HuscFv). The plate was kept at 4 °C overnight, and the wells were washed with PBS containing 0.05% Tween-20 (PBST). Rabbit anti-E-tag antibody (1:5000, Abcam) was added to each well (100 μL/well) and the plate was kept at 37 °C for 3 h. After washing all wells with PBST, horseradish peroxidase (HRP)-conjugated anti-rabbit Ig antibody (1:5000, SouthernBiotech, Birmingham, AL, USA) was added to each well (100 μL/well), kept at 37 °C for 1 h, washed, and then the ABTS substrate (100 μL/well, Seracare, Milford, MA, USA) was added to each well and incubated at room temperature (RT, 25 ± 2 °C) for 1 h. The optical density at 405 nm of the content in each well was determined against the blank (wells added with PBS instead of the *E. coli* lysate). To test the binding of HuscFvs to the human EcOX40 expressed on the HEK-OX40 cells, *E. coli* lysates containing HuscFvs were added to HEK293T cells and HEK-OX40 cells contained in separate tubes (5 × 10^5^ cells in 100 μL of 5× FACS buffer, i.e., cold 10% FBS-PBS-0.02% NaN_3_/tube), and the tubes were placed on a rotator at 4 °C for 1 h. The cells were washed with the FACS buffer three times and resuspended in 25 μL of the same buffer. Rabbit anti-E-tag antibody (1:500, Abcam) was added to the cells in each tube (25 μL/tube) and the tubes were placed on ice for 1 h. The cells were washed three times and resuspended in 12.5 μL FACS buffer. An Alexa Fluor 647-conjugated goat anti-rabbit Ig antibody (12.5 μL, 1:200, Invitrogen) was added to the cells and the tubes were placed on ice for 1 h. The cells were washed with FACS buffer, resuspended in 500 μL of 1% paraformaldehyde in PBS, and subjected to flow cytometric analysis using a flow cytometer (BD LSRFortessa™ Cell Analyzer, BD biosciences).

Complementarity-determining regions (CDRs) and immunoglobulin framework regions (FRs) of the HuscFvs were determined. *Escherichia coli* clones that expressed HuscFvs bound to human EcOX40 detected by both indirect ELISA and flow cytometry were grown in 5 mL LB-A broth at 37 °C with 250 rpm shaking aeration overnight. The bacterial cells were harvested by centrifugation (4000× *g*, 4 °C, 30 min) and the phagemids they contained were extracted by using a plasmid mini-extraction kit (Favogen, Ping-Tung, Taiwan). The isolated phagemid DNAs were Sanger-sequenced (1st Base). The nucleotide sequences were subjected to the CLC Main Workbench 8 program (Qiagen, Hilden, Germany); the nucleotide sequences of *huscfvs* of individual *E. coli* clones were deduced; and the CDRs and FRs of both VH and VL domains were analyzed using the IMGT/V-QUEST analysis tool, the International Immunogenetics Information, IMGT Systems (https://academic.oup.com/nar/article/50/D1/D1262/6455007?login=false, accessed on 1 November 2019) [47]. The *huscfv* sequences of all the *E. coli* clones were multiply aligned by using Clustal Omega [48] and compared.

### 2.7. Homology Modeling and Intermolecular Docking to Predict the OX40 Ectodomain Sites Bound by the HuscFvs

The three-dimensional (3D) structures of the HuscFvs were modeled by subjecting the HuscFv amino acid sequences to the online I-TASSER server [49]. The geometric and physical qualities of the I-TASSER-predicted 3D structures were improved by making them become closer to their native state using the ModRefinder [50] and the Fragment-Guided Molecular Dynamics (FG-MD) [51] systems, respectively. For the human EcOX40 homology model, the protein crystal structure PDB 2HEV [52] which was built from amino acids 29-214 of human OX40 (NP_003318.1) was used as the template for the I-TASSER modeling of the full-length EcOX40 protein. The human EcOX40 model derived by I-TASSER was also refined by the ModRefinder and FG-MD systems. For intermolecular docking, the modeled HuscFvs were docked against the modeled human EcOX40 by using the ClusPro Protein Docking server [53]. The largest cluster size with minimal local energy and a near-native state of the protein conformations was chosen from about 30 models derived from the ClusPro for each docking. The protein structural models and the molecular interactions were built and visualized by using the Pymol software (The PyMOL Molecular Graphics System, Version 1.3r1edu, Schrodinger, LLC, New York, NY, USA) and BioVia (Discovery studio). Interactive residues and chemical bonds between the HuscFvs and the human EcOX40 were determined. 

### 2.8. Large-Scaled Production of 6× His-Tagged-HuscFvs to EcOX40 

The *huscfvs* coding for HuscFvs to EcOX40 were subcloned from pCANTAB5E phagemids into the pET23b+ expression vector and transformed into DH5α competent *E. coli*. Bacterial colonies carrying the recombinant *6× his-huscfv-*pET23b+ plasmids were screened by direct colony PCR, and positive clones were cultured in 10 mL of LB broth containing 100 μg/mL ampicillin (LB-A) broth. The plasmids were extracted for DNA sequencing. The verified recombinant plasmids were transformed into competent BL21 (DE3) *E. coli* expression hosts. One positive transformant of each original clone was grown in LB-A broth at 37 °C with shaking aeration (250 rpm) overnight. The overnight culture (100 μL) was added to 200 mL of fresh Lennox broth containing 100 μg/mL ampicillin (LN-A) and incubated with shaking at 250 rpm until the OD at 600 nm reached 0.3–0.5. Isopropyl β-d-1-thiogalactopyranoside (IPTG) was added to the culture (0.1 mM final concentration) and the culture was incubated further for 5 h. The bacterial cells collected from each culture after centrifugation (4500× *g*, 4 °C, 15 min) were resuspended in 1 mL PBS and homogenized by sonication (30% amplitude) in an ice bath for 1 min. The clear supernatants were collected after centrifugation (12,000× *g*, 4 °C, 20 min). The HuscFvs in the supernatants were purified by using the TALON® Metal Affinity resin (Clontech Lab, Mountain View, CA, USA (presently Takara Bio USA)) and eluted with 150 mM imidazole solution into 1 mL fractions. The purified 6× His-tagged HuscFvs to EcOX40 were subjected to SDS-PAGE and Coomassie Brilliant Blue G250 (CBB) staining and Western blot analysis for verification.

### 2.9. Production of Human Bivalent HuscFv-Fcγ Antibodies to EcOX40

DNA sequences coding for HuscFvs (*huscfvs*) of the selected *E. coli* clones were codon-optimized for their expression in mammalian cells. The *huscfvs* were analyzed for their codon adaptation index (CAI) and the nucleotides were changed by using the vector NTI software (Thermo Fisher Scientific) and the vector builder [54]. The codon-optimized *huscfv* sequences with *Sac*I and *Bgl*II endonuclease restriction sites (for subsequent cloning) were synthesized (IDT, Coralville, IA, USA). 

The *huIgG1 Fc2*-pINFUSE mammalian expression vector (Invivogen) was transformed into JM109 *E. coli* by heat-shock transformation; the transformed bacteria were spread onto an LB-Zeocin agar plate, and the plate was incubated at 37 °C overnight. Bacterial colonies carrying the recombinant *huIgG1 Fc2*-pINFUSE plasmids were screened by direct colony PCR using the HTL5UTR forward and EBV-RP reverse oligonucleotide primers. The plasmids from the PCR-positive *E. coli* clones were extracted using a plasmid extraction kit (GeneAid, New Taipei, Taiwan). The bacterial clones with the verified plasmids were grown individually in LB-Zeocin broth at 37 °C for 16 h and the plasmids were extracted from the bacteria. The codon-optimized *huscfvs* were digested with the *Sac*I and *Bgl*II restriction enzymes and ligated to the similarly cut *huIgG1 Fc2*-pINFUSE plasmids vectors using T4 ligase. The recombinant plasmids were transformed into JM109 *E. coli,* and the transformed *E. coli* clones harboring plasmids with the inserted *huscfvs* were screened again via direct colony PCR as above. Positive clones were grown for plasmid extraction using a plasmid extraction kit (Favogen) and sequenced (1st Base). *Escherichia coli* clones with verified plasmids were grown and their plasmids were extracted and purified by using the EndoFree Maxi Plasmid kit (Tiangen Biotech)

For the production of human bivalent HuscFv-Fcγ antibodies (designated fusion antibodies), HEK293E suspension cells in IBA solution (IBA suspensions life sciences, Göttingen, Germany) were cultured in MEXi-CM serum-free medium (IBA) in Erlenmeyer culture flasks (Corning, Glendale, AZ, USA) at 37 °C in a 5% CO_2_ atmosphere with shaking aeration (125 rpm). The MEXi-CM serum-supplemented culture medium was gradually replaced with CDM4HEK293 serum-free medium (Cytiva) until the cells could grow well in the latter alone, which took 3–4 weeks. Log-phase-grown HEK293E cells were suspended in SFM4Transfx-293 transfection medium (Cytiva) (1 × 10^6^ cells/mL); the cells were incubated overnight as above, and they were ready for transfection the next day. For the cell transfection, the *huscfv-huIgG1 Fc2*-pINFUSE vector was mixed with a transfection reagent (FectoPro, PolyPlus transfection, Illkirch, France). The ratio of plasmid DNA/transfection reagent/booster reagent was 1:1.5:1; the mixture was added to the HEK293E suspension cells, and the preparation was incubated as above for 4 h. The CDM4HEK293 medium was added, and the cells were incubated further for 7–10 days or until 60–75% of the cells were dead. The cell-spent medium containing the fusion antibodies was harvested. The fusion antibodies were purified by mixing the cell culture supernatant containing the fusion antibodies with 10× IgG binding buffer (200 mM sodium phosphate buffer, pH 7.0), filtered through a 0.45-μm membrane, and the preparations were passed through protein G column chromatography (Cytiva). Fractions (1 mL) were collected (ÄKTAprime, Cytiva) and the presence of the fusion antibodies was determined by SDS-PAGE and CBB staining.

### 2.10. Effective Concentration 50 (EC50) and Binding of the HuscFvs and Bivalent HuscFv-Fcγ Antibodies (Fusion Antibodies) to Human OX40

An indirect ELISA was performed to determine the effective concentration 50 (EC50) of the fusion antibodies (bivalent HuscFv-Fcγ) for purified EcOX40 from HEK293E suspension cells. 

Binding of the fusion antibodies to OX40 on activated human T cells was performed using flow cytometry. 

Co-immunoprecipitations using HEK-OX40 cells and flow cytometric analysis were performed to demonstrate binding of the HuscFvs and fusion antibodies to full-length human OX40. 

For the indirect ELISA, purified EcOX40 from the culture supernatant of HEK293E suspension cells was dissolved in PBS (5 μg/mL). Next, 100 μL aliquots were added to each well of the EIA/RIA plate (Corning) and the plate was kept at 4 °C overnight. The coated wells were washed with PBS, and each well was blocked with 200 μL of 5% skim milk in PBS at 37 °C for 1 h. After washing, various concentrations of fusion antibodies (0.3125, 0.625, 1.25, 2.5, 5, 10, 20 and 40 μg/mL) in 100 μL PBS were added separately to the EcOX40-coated wells, kept at 37 °C for 3 h, washed with PBST, and 100 μL of horseradish peroxidase (HRP)-conjugated anti-human IgG Fc antibody (1:5000, Invitrogen) was added to each well. The plate was incubated at 37 °C for 1 h, washed, and 100 μL of the ABTS substrate (Seracare, Milford, MA, USA) was added to each well. The optical density at 405 nm of the content in each well was determined against blank (wells added with buffer instead of fusion antibodies) using the Synergy H1 Hybrid Multi-Mode Microplate Reader (BioTek, Santa Clara, CA, USA). The effective concentration 50 (EC50) of the fusion antibodies was calculated. 

The binding of the fusion antibodies to human EcOX40 on human T cells was also determined by flow cytometric analysis. Human T cells were isolated from PBMCs by magnetic-based negative selection using the EasySep^TM^ human T cell enrichment kit (Stemcell technologies, Vancouver, BC, Canada). The T cells were cultured with 0.5 μg/mL of PHA-L mitogen for 3 days to stimulate OX40 expression. Activated T cells were collected, washed with PBS, resuspended at 1 × 10^5^ cells in 25 μL of 20% FBS-PBS-0.02% NaN_3_ (blocking buffer), and placed on ice for 30 min. After that, 25 μL of 40 μg/mL of fusion antibodies were added to the cells, placed on ice for 30 min, and then washed with 2% FBS-PBS-0.02% NaN_3_ (FACS buffer) three times. The cells were counter-stained with mouse anti-OX40-Alexa Fluor 647 conjugate and anti-human IgG Fc-Brilliant violet 405 conjugate and placed on ice for 30 min. The stained cells were then washed with FACS buffer, fixed with 1% paraformaldehyde in PBS, and assessed for EcOX40 binding of the fusion antibodies by flow cytometry (BD LSRFortessa, BD biosciences).

Binding of the HuscFvs/fusion antibodies to human OX40 was determined via a co-immunoprecipitation assay. The HEK-OX40 cells (5 × 10^6^ cells) that expressed human OX40 were suspended in 500 μL of PBS, 2 μg of HuscFvs (6× His-tagged-HuscFvs) or fusion antibodies were added, and the cells were placed on a rocker at 4 °C for 3 h. The cells were washed with cold PBS five times. The cell pellets were then lysed with 500 μL of 1% Triton-X 100-Tris lysis buffer containing a protease inhibitor cocktail (Sigma-Aldrich) on ice for 1 h and centrifuged (15,000× *g*, 4 °C, 30 min). The clear cell lysate was collected, mixed with 2 μg of anti-6× His antibody and placed on a rocker at 4 °C for 3 h. Two microliters of protein G magnetic beads (Genscript) were added into the mixture and placed on the rocker at 4 °C for 1 h. The mixture-containing tube was placed on a magnetic stand and the beads were washed thoroughly with 1 mL of cold 1% Triton-X 100 Tris lysis buffer. The proteins were eluted from the beads by adding reducing sample buffer, boiled for 5 min, and subjected to SDS-PAGE and Western blotting. Proteins coprecipitated with OX40 were detected by using the rabbit anti-human OX40 antibody (clone BLR042F, 1:5000, Abcam) in the Western blot analysis.

### 2.11. NF-κB Assay for Determining the Ability of HuscFvs to EcOX40 to Stimulate OX40 Signaling 

Preliminary experiments were performed by using the NF-κB assay to select HuscFvs that have OX40-stimulatory potential. 

HEK-OX40 cells were seeded into a well of a 6-well-culture plate (5 × 10^5^ cells/well) and incubated at 37 °C in a 5% CO_2_ incubator overnight. The lipofectamine™ 3000 transfection reagent (Thermo Fisher Scientific) was prepared: in one test tube, 1 μL of P3000TM, 25 μL of Opti-MEM™I reduced serum, and 1 μg of pNL3.2.NF-κB-RE[*NlucP*/NF-κB-RE/Hygro] vector (Promega, Madison, WI, USA) were added and mixed; in another test tube, 1.5 μL of Lipo3000 was mixed with 23.5 μL of Opti-MEM™ I reduced serum. Thereafter, the contents of the two test tubes were mixed well, and kept at 25 °C for 15 min. The mixture was gently added dropwise to the HEK-OX40 cells in the well and the cells were incubated at 37 °C in a humidified CO_2_ incubator for 72 h. The HEK-OX40 cells that contained the pNL3.2.NF-κB-RE[*NlucP*/NF-κB-RE/Hygro] plasmids were designated “HEK-OX40 reporter cells”. 

For the NF-κB assay, HEK-OX40 reporter cells were seeded into wells of a 96-well white plate (15,000 cells/well) and incubated at 37 °C in a 5% CO_2_ incubator overnight. Various concentrations of purified 6× His-tagged HuscFvs (5, 10, 20 and 40 μg/mL) were added to the HEK-OX40 reporter cells. Mouse anti-His antibody (at half the concentration of the added 6× His-tagged HuscFvs) was added into each well to cross-link the HuscFvs that bound to EcOX40 on the cells to induce OX40 ligation/clustering. The HEK-OX40 reporter cells incubated with mouse anti-His antibody (Biolegend) served as the background control (no HuscFvs/crosslinked HuscFvs); HEK-OX40 reporter cells incubated with 2 ng/mL of human TNF-α served as the positive control for the NF-κB reporter system. After 6 h of treatments, 100 μL of the Nano-Glo substrate (Promega) was added into each well, and luciferase activity was measured by using the BioTek Synergy H1 Multimode Reader (BioTek, Santa Clara, CA, USA). 

### 2.12. Preparation of Primary T Cells 

Human PBMCs were isolated from the buffy coat of heparinized whole blood after Ficoll-Hypaque (Histopaque-1077; Sigma-Aldrich) density-gradient centrifugation. The PBMCs were washed twice with Dulbecco’s phosphate-buffered saline (DPBS), counted, and resuspended in 2% FBS-1 mM EDTA-DPBS (sorting medium) at 5 × 10^7^ cells/mL. T cells were isolated from the PBMCs by magnetic-based negative selection using the EasySep^TM^ human T cell enrichment kit (Stemcell technologies) according to the manufacturer’s protocol. The purity of the T cells (>95%) was checked by staining with anti-CD3-APC/Cy7 (clone SK7, Biolegend). The T cells were then resuspended in 10% FBS-AIM-V medium (Life Technologies, Thermo Fisher Scientific, Carlsbad, CA, USA).

### 2.13. T Cell Proliferation Assay 

T cells (1 × 10^6^ cells/mL in DPBS) were labeled with 5 μM of CellTrace Violet (Life Technologies) and incubated at 37 °C for 20 min. Medium containing 2% FBS was added (five times the original volume of the staining medium) and centrifuged once to remove free dye. The cell pellet was resuspended in 10% FBS-AIM-V medium (Life Technologies) at 1 × 10^6^ cells/mL, and 100 μL aliquots of the cells were added into wells precoated with 100 μL of anti-CD3 antibody (0.5 μg/mL, clone UCHT-1, Biolegend) and fusion antibodies to OX40 (0.25, 0.5, 1.0 and 2.0 μg/mL); soluble anti-CD28 (1 μg/mL; clone CD28.2, Biolegend) was added to the wells containing T cells. T cells in culture medium alone served as the non-stimulated cell control. The cells were then kept at 37 °C in a CO_2_ incubator for 5 days. They were then harvested and the reduction of CellTrace Violet (indicating cell proliferation) was assessed via flow cytometry (BD FACSymphony A1, BD biosciences).

### 2.14. Detection of Intracellular Cytokines and Granzyme B

Human T cells in 10% FBS-AIM-V medium (1 × 10^6^ cells/mL) were stimulated with immobilized anti-CD3 antibody and fusion antibodies and soluble anti-CD28 as above for 2 days. The cytokine blocker Brefeldin A (Biolegend) was added to the cells in each well 12 h prior to measuring the intracellular cytokines and granzyme B. The total T cells (CD3^+^ cells) were also stained for CD4^+^ and CD8^+^ subpopulations. In brief, 1 × 10^6^ cells were resuspended in 50 μL of 10% human AB serum-FACS buffer and placed on ice for 30 min. The cells were stained with a cocktail of antibodies (Biolegend) against CD3-APC/Cy7 (clone SK7), CD4-Alexa Fluor 488 (clone SK3), and CD8-PE (clone SK1) and placed on ice for 30 min. After washing, the stained cells (CD3^+^ T cell and CD4^+^ and CD8^+^ T subpopulations) were then fixed with 4% paraformaldehyde-DPBS at RT for 15 min, washed twice with DPBS, and incubated in intracellular staining permeabilization wash buffer (Biolegend) for 20 min at RT. Cytokine/granzyme B-specific antibodies (Biolegend), i.e., anti-IL-2-Brilliant Violet 421 (clone MQ1-17H12), anti-IFN-γ-Brilliant Violet 421 (clone 4S.B3), anti-TNF-α-Brilliant Violet 421 (clone Mab11) or anti-granzyme B-Alexa Fluor 647 (clone QA16A02) were added to the permeabilized cells and incubated at RT for 30 min. The stained cells were washed with intracellular staining permeabilization wash buffer and the presence of intracellular cytokines/granzyme B was assessed by flow cytometry (BD FACSymphony A1, BD biosciences).

### 2.15. Determination of the Enhancing Effect of the Fusion Antibodies on T Cell Survival 

To determine the effect of fusion antibodies on T cell survival, T cells (1 × 10^6^ cells/mL of 10% FBS-AIM-V medium) that had been activated with immobilized anti-CD3 antibody and fusion antibodies in the presence of soluble anti-CD28 were cultured at 37 °C in CO_2_ incubator for 5 days. The cells were harvested, washed twice with DPBS, then stained with 0.5 µM ethidium homodimer D-1 (EthD-1) (Invitrogen) and 400 nM Apotracker green (Biolegend). Percentages of living cells, total dead cells, necrotic cells, and early/late apoptotic cells were assessed via flow cytometry (BD FACSymphony A1, BD biosciences) and compared with the CD3-CD28-activated T cells in the medium alone (no immobilized fusion antibodies)

### 2.16. Preparation of Monocyte-Derived Dendritic Cells (MoDCs) 

Human PBMCs were suspended in sorting medium (2% FBS-1 mM EDTA-DPBS) at 5 × 10^7^ cells/mL. Monocytes were isolated from the PBMCs by magnetic-based negative selection using the EasySep Human Monocyte Enrichment kit (Stemcell technologies) according to the manufacturer’s protocol. The purity of the monocytes was checked by staining them with anti-CD14-fluorescein isothiocyanate (FITC) (clone QA19A47, Biolegend). The purity of CD14-positive cells was higher than 90%. Monocytes were then resuspended in 10% FBS-AIM-V medium (Life Technologies) at 1 × 10^6^ cells/mL. The monocytes (500 μL) were cultured in 24-well flat-bottomed tissue culture plates. For the differentiation of monocytes into moDCs, the cell culture medium was supplemented with 800 units/mL of recombinant human GM-CSF and 400 units/mL of recombinant human IL-4 (Stemcell technologies) (differentiation medium). Cell-spent differentiation medium was replaced with 50% fresh differentiation medium on day 3 and cultured further for 2 more days. Thereafter, the cells were cultured in maturation medium (medium supplemented with 400 ng/mL CD40L (Biolegend) and 100 units/mL TNF-α (Stemcell technologies)) for 2 days [55,56]. The cells were harvested and checked for expression of the immunophenotypic markers, CD14 and CD83, by staining with anti-CD14-FITC and anti-CD83-Alexa Fluor 647 (clone HB15e, Biolegend) and analysis via flow cytometry (BD FACSymphony A1, BD biosciences).

### 2.17. Preparation of Tumor Cell Lysate

The ovarian cancer cells, SK-OV-3, from a confluent culture were harvested and washed twice with PBS. The cells were resuspended to a density of 1 × 10^7^ cells/mL in sterile PBS and kept in a water bath at 42 °C for 1 h and 37 °C for 2 h. Then, they were lysed by rapid freezing in liquid nitrogen and thawing in a 37 °C water bath for 4–5 cycles [57]. The preparation was centrifuged (10,000× *g*, 4°C, 30 min), and the clear lysates were collected and passed through a 0.2 μm filter. The protein concentration of the lysate was determined using the BCA protein assay (Bio-Rad, Munich, Germany).

### 2.18. Preparation of Tumor-Lysate-Pulsed DCs and Tumor-Antigen-Primed T Cells

Monocyte-derived DCs (1 × 10^5^ cells) in 1 mL 10% FBS-AIM medium were added to the ovarian cancer cell lysate (100 μg/mL) and incubated at 37 °C in aCO_2_ incubator for 12 h. MHC class-I partially matched human T cells were added to the tumor-lysate-pulsed DCs at a ratio of 5:1, and the cells were cultured for 7 days. On days 3 and 5, 50% of the spent culture medium was replaced with fresh medium containing low doses of 10 units/mL IL-2 (Stemcell technologies) and 5 ng/mL IL-7 (Stemcell technologies). On day 5, the cells were observed under an inverted microscope, and sample was taken to check for the expression of OX40 by staining with an antibody cocktail containing anti-CD3-APC/Cy7, anti-CD4-Alexa Fluor 488, anti-CD8-PE/Dazzle 594, and anti-OX40-Alexa Fluor 647, followed by analysis via flow cytometry. On day 7, the tumor-antigen-primed T cells were harvested, washed twice with PBS, and resuspended in fresh 10% FBS-AIM-V medium (1 × 10^6^ cells/mL). 

### 2.19. Cytotoxicity Assay

After stimulation with the tumor-lysate-pulsed DCs, the tumor-antigen-primed T cells were additionally stimulated by incubating with anti-EcOX40 fusion antibodies for 24 h. The calcein-release assay was used for determining the ovarian tumor cell lysis caused by the fusion-antibody-stimulated tumor-antigen-primed T cells in comparison with the tumor-antigen-primed T cells without fusion antibody stimulation (control tumor antigen primed-T cells). Ovarian cancer cells were labeled with 1 μM of calcein-AM (Invitrogen), and 1 × 10^4^ labeled cells were seeded into individual wells of a black 96-wellplate for 12 h. The fusion-antibody-stimulated tumor-antigen-primed T cells and control cells were added separately to the calcein-labeled ovarian cancer cells at target/effector (T:E) ratios of 1:5, 1:10, and 1:20. The cells of all treatments were cultured at 37 °C in CO_2_ incubator for 24 h. After incubation, 100 μL of the supernatant from each well (triplicate wells for each treatment) was collected, and the fluorescence intensity of the released calcein was measured using the BioTek Synergy H1 Multimode Reader (BioTek). Two independent and reproducible experiments were performed using T cells of two blood donors whose MHC class-I partially matched that of the ovarian cancer cells. Specific lysis of the cancer cells was calculated as follows: Specific calcein-release = (experimental fluorescence intensity − spontaneously released fluorescence intensity) ÷ (maximal fluorescence intensity − spontaneously released fluorescence intensity) × 100 

### 2.20. Statistical Analysis

Data obtained from independent experiments were analyzed using Kruskal–Wallis and Dunn’s multiple comparison tests using GraphPad Prism, version 10 (GraphPad Software Inc., San Diego, CA, USA). Two-tailed *p* values of less than 0.05 were considered significant (*, *p* < 0.05; **, *p* < 0.01).

## 3. Results

### 3.1. OX40-Overexpressed Cells (HEK-OX40 Cells)

Total RNA extracted from PHAL-activated human PBMCs that expressed high levels of surface OX40 (Figure 1A) was reverse transcribed to cDNA, and the cDNA was used as a template for PCR amplification of the full-length human OX40 gene (human *tnfrsf4*) (Figure 1B). The *tnfrsf4* was ligated with the bicistronic *P2A*-*ZsGreen1*-pLVX-puro vector (Figure 1C). The recombinant *tnfrsf4*-*P2A*-*ZsGreen1*-pLVX-puro vector and the Lenti-X packaging single shots (VSV-G) were co-introduced into HEK293T cells. Lentivirus particles in the cell culture supernatant were used to infect HEK293T cells to produce cells that expressed ZsGreen1 fluorescence (Figure 1D) and overexpressed EcOX40 on the cell surface (HEK-OX40 cells). Up to 98.8% of the lentivirus-infected cells expressed OX40 on their surface (Figure 1E). Expression of the ZsGreen1 protein by the cells correlates with OX40 protein expression by the cells. 

### 3.2. Phage Panning and the Generation of Phage-Transformed E. coli Clones That Produced EcOX40-Bound HuscFvs 

A recombinant ectodomain of human OX40 (EcOX40) with inherent OX40-ligand-binding activity (Abcam) and HEK-OX40 cells were used as antigens in phage-panning to select HuscFv-displaying phage clones that bound to EcOX40 from the HuscFv phage display library. The antigen-bound phages were used to infect HB2151 *E. coli*, and the phage-infected *E. coli* clones were screened for *huscfvs*. Amplicons of *huscfvs* from the *E. coli* clones carrying *huscfv*-phagemids are shown in Figure 2A. The *huscfv*-positive *E. coli* clones were grown and induced by IPTG to produce soluble HuscFvs. Figure 2B shows HuscFv proteins in the lysates of *huscfv*-positive HB2151 *E. coli* clones. HuscFvs in the lysates of 18 *E. coli* clones derived from panning with recombinant EcOX40 gave positive signals against EcOX40 in the indirect ELISA. These were clones B1, C2, G2, A3, D3, B4, C4, H4, H5, B8, F8, D9, E9, A10, G10, D11, F11, and H11 (Appendix A). From phage-panning using HEK-OX40 cells as a panning antigen, 15 phage-infected *E. coli* clones gave positive ELISA binding to the EcOX40 protein (clones C4, C5, C8, C26, C28, C33, C36, C39, C41, C42, C58, C60, C61, C62, and C78 (Appendix A)). 

The EcOX4-bound HuscFvs in the lysates of the 33 phage-infected HB2151 *E. coli* clones were then tested for binding to EcOX40 on the surface of the HEK-OX40 cells by flow cytometry using the original HB2151 *E. coli* lysate (HB lysate) as the background (negative-binding control) and the anti-OX40 antibody (Biolegend) as the positive-binding control. HuscFvs of 12 clones (C2, D11, F8, C28, C36, C39, C41, C42, C60, C61, C62, and C78) were bound to EcOX40 on the HEK-OX40 cells (Figure 2C); these HuscFvs did not bind to HEK293T cells (Figure 2D). HuscFvs of the other 21 *E. coli* clones did not bind to HEK-OX40 cells or bound non-specifically to HEK293T cells; these were not tested further.

The *huscfv* sequences coding for the HuscFvs of the 12 HB2151 *E. coli* clones that bound to both recombinant EcOX40 and surface-expressed EcOX40 on the HEK-OX40 cells were analyzed by using the IMGT/V-QUEST analysis tool. There were five for which sequences were the same, indicating that they were sibling clones. Thus, only clones C2, D11, F8, C36, C41, C62, and C78 were tested further. The deduced amino acid sequences of the HuscFvs produced by these seven clones were compared by using the Clustal Omega program, which showed that the amino acid sequences in their CDR and FR regions are different. Their FR sequences showed >90% identity to human immunoglobulin FRs, based on the International Immunogenetics Information System (Appendix A); thus, the HuscFvs are human proteins. 

### 3.3. Prediction of the EcOX40 Sites Bound by the HuscFvs

Homology modeling and intermolecular docking were performed to predict the sites of EcOX40 that formed the contact interface with the HuscFvs. Three-dimensional (3D) models of HuscFvs of the seven *E. coli* clones (C2, D11, F8, C36, C41, C61, and C78) were built by using I-TASSER and refined by ModRefiner and FG-MD tools. The EcOX40 model was from the Zhang Lab [51]. From the Ramachandran plots, both the full-length EcOX40 3D (Figure 3A) and the HuscFv 3D (represented in Figure 3B) models show >90% of the most favored regions. The reliability of the 3D models of EcOX40 and HuscFvs of the *E. coli* clones are detailed in Appendix A. 

Figure 3C–I illustrate the contact interfaces between EcOX40 and the HuscFvs of individual *E. coli* clones. Details of interactive residues and the chemical bonds involved are shown in Appendix A. From the intermolecular docking, the HuscFvs of clones C2, D11, F8, C36, C41, and C78 interacted with the cysteine-rich domain-2 (CRD2) of EcOX40, which has been shown to function in OX40 stimulation, while the HuscFv of clone C62 formed a contact interface with CRD4 and the stalk of the EcOX40, which are not involved in OX40 stimulation [45]. Thus, the HuscFvs of clones C2, D11, F8, C36, C41, and C78 were selected for further experiments.

For the large-scale production and purification of the HuscFvs, the *huscfvs* of HB2151 *E. coli* clones C2, D11, F8, C36, C41, and C78 were ligated to pET23b+ and the *huscfv*-pET23b+ vectors were introduced separately into BL21 (DE3) *E. coli*. The transformed bacteria carrying *huscfv*-pET23b+ vectors (Figure 4A) were grown under IPTG induction to produce 6× His-tagged HuscFvs. Clones F8 and C78 produced HuscFvs in small amounts. The HuscFvs of clones C2, D11, C36, and C41 were purified by the Talon metal-affinity resin (Figure 4B,C) and were tested further.

### 3.4. Binding of HuscFvs to OX40 

The purified 6× His-tagged-HuscFvs of clones C2, D11, C36, and C41 were then tested for their binding to human OX40 in the HEK-OX40 lysate by co-immunoprecipitation. The 6× His-tagged-HuscFvs of individual *E. coli* clones were mixed with the HEK-OX40 cell lysate. The HuscFv-EcOX40 complexes were pulled down by mouse protein G magnetic beads precoated with anti-His antibodies. The immunoprecipitated proteins were eluted from the beads and subjected to Western blot analysis using a rabbit anti-OX40 antibody. As shown in Figure 4D, the HuscFvs of the four *E. coli* clones (~26–34 kDa) could bind to human OX40 protein in the HEK-OX40 cells (~50 kDa) and the two parties were coprecipitated by the anti-His coated protein G beads. 

### 3.5. HuscFvs to EcOX40 Mediate Activation of the NF-κB Transcription Factor

To investigate the effect of HuscFvs to OX40 on the activation of OX40 signaling, the NF–κB assay was performed. HEK-OX40 cells that contained the NF-κB reporter system were prepared. HEK-OX40 cells were transiently transfected with pNL3.2.NF-κB-RE[*NlucP*/NF-κB-RE/Hygro] vector to generate the HEK-OX40-NF-κB reporter cells (HEK-OX40 reporter cells). The purified 6× His-tagged-HuscFvs of individual *E. coli* clones (HuscFvC2, HuscFvD11, HuscFvC36, and HuscFvC41) were added to the HEK-OX40 reporter cells (100 μL of 5, 10, 20, and 40 μg/mL) and the OX40-bound HuscFvs were cross-linked by adding mouse anti-His antibody to induce multimerization of the OX40 protein (diagram in Figure 4E). The results demonstrated that HuscFvC2 and HuscFvC41 could induce the ligation of OX40, which led to NF-κB activation in a dose-dependent manner (Figure 4F). In contrast with the HuscFvD11- and HuscFvC36-treated cells, the luciferase activities were not altered when compared to an anti-His background (no HuscFvs). The results indicate that the crosslinked HuscFvC2 and HuscFvC41 could induce OX40 signaling leading to NF-κB activation; thus, these HuscFvs were studied further.

### 3.6. Bivalent HuscFv-Fcγ to EcOX40 (Fusion Antibodies) 

Because the crosslinked HuscFvC2 and HuscFvC41 stimulated OX40 on HEK-OX40 reporter cells to increase NF-κB activity, these HuscFvs were used for the generation of bivalent HuscFv-Fcγ antibodies (fusion antibodies). For this experiment, gene sequences coding for HuscFvs (*huscfvs*) of the *E. coli* clones C2 and C41 were codon-optimized (Appendix A) for their expression in mammalian cells. The optimized genes were separately ligated to *huIgG1 Fc2*-pINFUSE plasmid vectors (Figure 5A). The *huscfv*-*huIgG1 Fc2*-pINFUSE vectors were transformed into the HEK293E suspension cells. The transformed cells were grown in CDM4HEK293 medium for 7–10 days and the fusion antibodies (diagram shown in Figure 5B) in the spent medium of each culture were purified by protein G column chromatography. Examples of the purified fusion antibodies are shown in Figure 5C–E.

### 3.7. Binding of Fusion Antibodies to OX40

The purified bivalent fusion antibodies, designated C2Fc and C41Fc, were tested for OX40-binding activity by co-immunoprecipitation (IP), flow cytometry, and indirect ELISA. 

For the co-immunoprecipitation, C2Fc and C41Fc were mixed with the HEK-OX40 cell lysate, and the immune complexes were pulled down by protein G magnetic beads, eluted from the beads, and subjected to SDS-PAGE and Western blot analysis. Bands of the fusion antibodies were detected by using a goat anti-huIgG (H + L) =HRP conjugate while bands of OX40 were detected using a rabbit anti-OX40 antibody. Bands of both OX40 (~50 kDa) and reduced fusion antibodies (~55 kDa) were found in the same eluate from the protein G magnetic beads in the Western blot analysis (Figure 6A). The Western blot pattern of OX40 in the HEK-OX40 cell lysate was used as the OX40 marker.

Both fusion antibodies were also tested for binding to EcOX40 on the HEK-OX40 cells and to native OX40 expressed on activated human T cells by flow cytometry. HEK-OX40 cells, constitutively expressing OX40 and ZsGreen1 green fluorescent protein, were stained with 20 μg/mL of fusion antibodies. The stained cells were counter-stained with anti-human IgG-Alexa Fluor 405 and the cell binding was measured via flow cytometry. Both C2Fc and C41Fc bound to HEK-OX40 cells (Figure 6B, upper panels) but did not bind to HEK293T wild-type cells (with no OX40 expression) (Figure 6B, lower panels). 

For testing the binding of the fusion antibodies to the native OX40, human T cells were stimulated with PHA-L for 3 days. The activated T cells were then mixed with C2Fc or C41Fc, followed by anti-human IgG-Alexa Fluor 405 and mouse anti-OX40-Alexa Fluor 647. The stained activated T cells were analyzed for dual-positive cells (OX40- and fusion antibody-stained cells) by flow cytometry. The results shown in Figure 6C demonstrate that C2Fc and C41Fc could bind to OX40 on activated T cells. 

The effective concentration 50 of the fusion antibodies, C2Fc and C41Fc, was determined using an indirect ELISA against EcOX40. Various concentrations of fusion antibodies (0.3125, 0.625, 1.25, 2.5, 5, 10, 20, and 40 μg/mL) were tested in the indirect ELISA for their binding to 0.5 μg/mL of EcOX40 (Abcam). The calculated EC50 values of C2Fc and C41Fc were 81.53 and 236.96 nM, respectively (Figure 6D,E).

### 3.8. Fusion Antibodies to EcOX40 Enhance the Proliferation, Cytokine/Granzyme B Production, and Survival of T Cells

To determine the effect of fusion antibodies to OX40 on T cell activities, assays for T cell proliferation, intracellular cytokines/granzyme B production, and T cell survival and death were performed.

To determine whether OX40 ligation by the fusion antibodies C2Fc and C41Fc was able to enhance the proliferation of OX40-expressing T cells, we measured the proliferation of T cells that had been activated by anti-CD3 and anti-CD28 to express surface OX40 (CD3-CD28-stimulated T cells) using the CellTrace violet assay. First, we titrated for the optimal concentration of the fusion antibodies for enhancing the proliferation of CD3-CD28-stimulated T cells. Normal human peripheral blood T cells were stimulated with immobilized anti-CD3 (UCHT-1) and different concentrations of fusion antibodies in the presence of soluble anti-CD28 (CD28.2). T cells stimulated only with immobilized anti-CD3 and soluble anti-CD28 served as a control. The wells of tissue culture plates were precoated with an anti-CD3 antibody and various concentrations of fusion antibodies. Human T cells were labeled with CellTrace Violet and added together with a soluble anti-CD28 antibody to the immobilized anti-CD3-coated wells with/without co-immobilized fusion antibodies. After 5 days, the cells were harvested and measured for the reduction of CellTrace violet (an indicator of cell-proliferation) by flow cytometry. Figure 7A shows histograms of unstimulated and CD3-CD28-stimulated T cells from flow cytometric analysis. Figure 7B,C are histograms showing the enhanced proliferation of CD3–CD28 stimulated T cells through various concentrations of C2Fc and C41Fc. High proliferation enhancement was obtained at 1 and 2 μg/mL of both fusion antibodies. Thus, the fusion antibodies were used at 1 μg/mL for testing the proliferation enhancement of peripheral blood T cells from four healthy blood donors. As shown in Figure 7D, C2Fc fusion antibodies enhanced the proliferation of CD3-CD28 stimulated T cells from all four blood donors. The C41Fc fusion antibodies only enhanced the proliferation of CD3-CD28-stimulated T cells from donors 3 and 4. The selected concentration (1 μg/mL) might be suboptimal for the C41Fc fusion antibodies, hence causing its lower activity in enhancing the proliferation of CD3-CD28-activated T cells of donors 1 and 2 compared to C2Fc. 

To determine the effect of C2Fc and C41Fc on cytokine and granzyme B production by activated T cells, we measured the levels of intracellular cytokines and granzyme B in stimulated CD3^+^ T cells, CD4^+^ T cells, and CD8^+^ T cells. T cells were stimulated with immobilized anti-CD3 and 1 μg/mL of C2Fc or C41Fc and soluble anti-CD28. After 2 days of stimulation, a cytokine blocker, Brefeldin A, was added for 16 h and the cells were then stained with anti-CD3, anti-CD4, and anti-CD8 and antibodies to intracellular cytokines (IL-2, IFN-γ, and TNF-α) and granzyme B. CD3–CD28-stimulated T cells in the medium served as controls. The results shown in Figure 8A–C,E–G,I–K demonstrate that C2Fc was able to increase the percentage of the cytokine-producing cells among the CD3^+^, CD4^+^, and CD8^+^ T cell populations, respectively, when compared to control stimulated cells in the medium alone (no fusion antibodies). However, the percentages of granzyme B-producing CD3^+^, CD4^+^, and CD8^+^ positive cells stimulated by C2Fc were not significantly different from the controls (*p* > 0.05), even though increasing trends were observed for all three T cell populations (Figure 8D,H,L). C41Fc significantly increased the percentage of granzyme B-producing cells in the CD3^+^ and CD4^+^ T cell populations (Figure 8D and Figure 8H, respectively), but only showed an increasing trend for intracellular cytokines in all three T cell populations, and for granzyme B only in the CD8^+^ population (*p* > 0.05). The representative dot plots of this experiment are shown in Figure 9.

The effects of the C2Fc and C41Fc fusion antibodies on T cell survival, T cell death, and apoptosis were determined via flow cytometry. T cells were stimulated with immobilized anti-CD3 and 1 μg/mL of C2Fc or C41Fc and soluble anti-CD28. After 5 days, the cells were stained with Apotracker Green (Apo-15 peptide, a calcium-independent probe for detecting apoptotic cells) and ethidium homodimer 1 (EthD-1). From the flow cytometric analysis, living cells, necrotic cells, early apoptotic cells, and late apoptotic cells were designated as Apotracker green− EthD-1−, Apotracker green− EthD-1+, Apotracker green+ EthD-1−, and Apotracker green+ EthD-1+, respectively (Figure 10A). The results demonstrate that C2Fc caused a significant increase in the percentage of living cells *(p* < 0.05) (Figure 10B) and significantly reduced the percentages of total cell death (*p* < 0.01) (Figure 10C) and early apoptotic cells (*p* < 0.001) (Figure 10E). Neither C2Fc nor C41Fc altered the percentage of late apoptotic cells (Figure 10F). C41Fc did not alter the percentages of living cells, total cell death, early, and late apoptotic cells but caused an increase in necrotic cells compared with control CD3-CD28-stimulated cells in the medium (*p* > 0.05) (orange bars, Figure 10A–F).

### 3.9. Fusion Antibodies to EcOX40 Enhanced the Cytotoxic Activity of Tumor-Antigen-Primed T Cells (Tumor-Reactive T Cells)

Experiments to determine whether fusion antibodies to EcOX40 could enhance the cytotoxic activity of effector T cells against ovarian cancer cells were performed. Peripheral blood mononuclear cells of healthy donors were screened for those with an MHC that matched with the SK-OV-3 ovarian cancer cells. The PBMCs of the MHC-class I partially matched donors were used for the generation of tumor-antigen-primed T cells (tumor-reactive T cells) which were then used as effector cells in the cancer-cell-killing experiments. To generate tumor-antigen-primed T cells, monocyte (Mo)-derived dendritic cells (DCs) were prepared. The MoDCs were pulsed with the ovarian cancer cell lysate, and the pulsed cells were used to prime T cells to generate the tumor-antigen-primed T cells.

Monocytes were isolated from the PBMCs of two healthy donors (donors X and Y). They were cultured in medium containing GM-CSF and IL-4 to promote their differentiation into immature dendritic cells (imDCs). The imDCs were then cultured in TNF-α and CD40L-containing medium for further differentiation into mature DCs (mDCs). The monocytes became bigger in size when they transformed into imDCs, and dendrites appeared when the imDCs became mature DCs (mDCs) (Figure 11A). The monocyte marker CD14 was downregulated as the monocytes differentiated into imDCs and mature DCs (Figure 11B). On the contrary, CD83 (DC marker) was upregulated in imDCs and mDCs compared with monocytes (Figure 11C).

A lysate of SK-OV-3 ovarian cancer cells was prepared and added to the mDCs. Autologous T cells were cocultured with the tumor-lysate-pulsed mDCs at a ratio of DCs:T cells of 1:5 (Figure 12A). After 5 days, the T cells divided, and large clusters of mDCs and T cells appeared (Figure 12A, right). After coculturing the tumor-lysate-pulsed mDCs and T cells for 5 days, the percentages of OX40-expressing T cells were increased by 29.1, 23.5, and 21.4% among total T cells, CD4^+^, and CD8^+^ T cells, respectively (Figure 12B). The tumor-antigen-primed T cells were incubated with 10 μg/mL of the C2Fc/C41Fc fusion antibodies or with medium alone (tumor-antigen-primed T cell control) for 2 days before they were cocultured with target cells, i.e., calcein-labeled-SK-OV-3 cells at a T:E ratio of 1:5, 1:10, or 1:20. The T-cell-mediated cytotoxicity to target ovarian cancer cells was determined by measuring the amount of calcein released from the dead cancer cells. The percentage of specific lysis was then calculated from the amount of the released calcein. The results in Figure 12C indicate that tumor-antigen-primed T cells exposed to C2Fc/C41Fc fusion antibodies had a trend of increasing ability in tumor-cell killing when the effector cell numbers were increased compared with tumor-antigen-primed T cells in medium control (no fusion antibodies). The data were from two reproducible experiments using two blood donors (donors X and Y). In each experiment, cells of each donor were tested in triplicate. Figure 13 illustrates the steps involved in the experiments for testing the ability of the fusion antibodies to enhance the anti-cancer activity of tumor-antigen-primed T cells. 

## 4. Discussion

An investigation into the role of the immune system in fighting against tumors was embarked on in the late 18th century [58,59]. During the early stages of tumor initiation, naïve T cells are primed and activated by tumor immunogenic antigens in the draining lymph nodes from where tumor-specific T cells recirculate back to destroy the cancer. Advanced cancers with highly infiltrated T cells into the TME have a good prognosis, i.e., enhanced survival [60]. Both CD4^+^ and CD8^+^ T cells are tumor destructors. Th1 cells secrete cytokines (IL-2, TNF-α, and IFN-γ) that stimulate M1 macrophages, NK cells, and CD8^+^ T cells to increase the respective anti-tumor activities and prolong cell survival [61]. Utilizing a cell–cell contact-dependent mechanism, the tumor-antigen-primed CD8^+^ T cells (CTLs) directly kill the target cancer cells through specialized exocytosis toward the immunological synapse to deliver (in a paracrine manner) death-inducing granules containing granzymes, perforin, cathepsin C, and granulysin. One CTL (as well as one NK cell) can kill multiple targets via different pathways of cytotoxicity (e.g., caspase-mediated apoptosis, cell membrane disruption, microtubule disruption, formation of GSDM pores in the membrane, and others) in a subsequential manner [62,63,64]. The CTLs also indirectly kill the tumor via the secretion of cytokines, especially IFN-γ [65]. Despite a multitude of opposing immunological factors (innate and adaptive), tumors manage to escape the host immune control and manifest clinically; they conduct this using several strategies, including the induction of an immunosuppressive TME; expression of immunosuppressive surface proteins, i.e., immune-checkpoint proteins (PD-1, CTLA-4, and LAG-3 that suppress effector cell activities) and ectonucleotidases (CD39/CD73 that convert ATP to immunosuppressive adenosine); downregulation of MHC-class I, FAS, and TRAIL to escape cell killing; secretion of immunosuppressive cytokines and molecules (IL-10, TGF-β, VEGF, idolamine-2,3-dioxygenase, lactate, prostaglandin-E2, galectin 1, and arginase); as well as by the creation of a hypoxic environment [66].

Insightful information on tumor immune suppressive factors and mechanisms has led to a cancer treatment revolution toward immunotherapy. Among the immunotherapeutic approaches, blocking the immune checkpoints has gained the most success in the treatment of many cancers. Nowadays, several immune-checkpoint blockers have been developed for therapeutic purposes [67]. Nevertheless, there are limitations in using the immune-checkpoint blockers. While a given blocker may be relatively effective for one cancer, it may fail to yield a beneficial outcome for other cancers [68]. Immune-checkpoint blocking alone is not sufficient to promote tumor regression in most patients [69]. In addition, repeated dosing with the checkpoint blockers may cause adverse immune-related reactions and toxicities to several organs/tissues, e.g., endocrine, cardiac, pulmonary, and gastro-intestinal [67]. 

Other than immune-checkpoint blockers, the stimulation of T cell costimulatory receptors, particularly tumor necrosis factor receptor (TNFR) superfamily members, e.g., OX40/CD134, 4-1BB, causes T cell proliferation, migration, survival, and activation, which can lead to tumor regression [70]. Co-receptor stimulators (agonists) can be used in combined immunotherapy with checkpoint blockers [71] or as an adjunctive, neo adjunctive, or induction therapy with conventional cancer treatments, i.e., surgery, radiation, and chemotherapy [72]. The proteins of the TNFR superfamily are generally characterized by an extracellular portion composed of several cysteine-rich domains (CRDs) for interacting with their natural ligands. The interaction of cell-surface TNFRSF proteins with their natural ligands leads to receptor homo-trimerization and clustering that triggers downstream signaling, including the activation of the nuclear factor-κB (NF-κB) and nuclear-factor-activated protein (NFAT) transcription factors, which drives immune cell activation, proliferation, and survival. For OX40, the extracellular portion (EcOX40) is composed of four CRDs (CRD1-CRD4) and the stalk. The natural ligand (OX40L) binds CRDs 1, 2, and 3 of the OX trimer to cause receptor clustering and activation of the NF-κB signaling cascade [73]. Agonistic monoclonal antibodies (mAbs) in a variety of formats that target T cell costimulatory receptors have been developed for cancer therapy [69]. Nevertheless, none of the OX40 agonists have reached the late phase of clinical trials [44]. Proper clustering of the naturally expressed, spatially distant OX40 molecules on the T cell surface such that the cytoplasmic tails of OX40 can recruit TRAFs (TRAF2 and TRAF5) to form a hexameric complex is required for adequate downstream signaling for sequential transcription factor and gene activation [44,70]. A hexameric OX40L-TRAF2-IgG4Fc fusion protein (MEDI6383) that clustered OX40 molecules by Fc gamma receptors (FcγRs) on the surface of adjacent FcγR-positive cells was found to stimulate and induce NF-κB activity in OX40-expressing T cells, as well as induce Th1 proliferation and cytokine production, and resistance to Treg-mediated suppression [39]. In this study, we generated novel OX40 agonistic antibodies in the form of bivalent fully human scFvs (HuscFvs) fused to a human IgG1 Fc fragment (fusion antibodies). The fusion antibodies by themselves could stimulate T cell proliferation, survival, and some effector functions, e.g., cytokine production, and they showed enhancement trends for other effector T cell activities like granzyme B production and the lysis of ovarian cancer cells that were used as a target cancer model. 

The production of fully human antibodies to human proteins (own species) requires specialized strategies. To generate the effective fusion antibodies to human OX40, we started by producing fully human single-chain antibody fragments (HuscFvs) that bound to the OX40 ectodomain (EcOX40). A phage library that contained phage particles displaying a repertoire of HuscFvs [45] was used as a tool for the production of the EcOX40-specific HuscFvs. The HuscFv phage display library that we used as a source of human OX40-specific HuscFvs in this study was generated from human immunoglobulin genes isolated from peripheral blood B cells of multiple healthy donors. All families and subfamilies of the genes coding for human VH (*vhs*) and VL (*vls*) domains were PCR-amplified using 48 pairs of degenerate oligonucleotide primers to *vhs* (16 forward and three reverse primers) and 26 pairs of degenerate primers to *vls* (13 forward and two reverse primers). The degenerate primers were chosen such that one *vh/vl* cDNA template could yield multiple variants after PCR amplification; this was to increase the antibody gene diversity and the possibility of also obtaining gene sequences that encode antibodies (VHs/VLs) to human self-antigens that are absent in the normal peripheral blood B cell pool (like immunoglobulin genes during in-frame V_H_-DJ_H_ rearrangements in pre-B cells and *IGK/IGL* gene rearrangements in small pre-B II cells prior to positive selection and allelic exclusion in immature B cells) [74]. The PCR-amplified *vh* and *vl* sequences were linked randomly via a polynucleotide linker to yield a repertoire of *vh-linker vl* or *huscfvs.* The *huscfv* repertoire was cloned into the pCANTAB5E phagemid downstream of the phage gene *p3*. The recombinant phagemids were introduced into competent TG1 *E. coli* and the phage-transformed *E. coli* were grown and co-infected with a M13KO7 helper phage. The complete phage particles displaying the human scFvs (HuscFvs) as fusion partners of the P3 protein and containing the respective HuscFv genes (*huscfvs*) in the phage genomes were recovered from the *E. coli* culture supernatant (i.e., the HuscFv phage display library was obtained) [45]. To select out phage clones displaying HuscFvs to the OX40 ectodomain (EcOX40), a commercialized recombinant EcOX40 that bound to OX40L and HEK-OX40 cells was used as a panning antigen. A total of 33 phage-transformed *E. coli* clones produced HuscFvs that bound to the recombinant EcOX40 via indirect ELISA, but among them, HuscFvs of 12 clones bound also to the surface-exposed OX40 on HEK-OX40 cells. These results verified that the HuscFv phage display library does contain phage particles that display HuscFvs to human self-antigens. The same library was used successfully for selecting out phage clones that displayed HuscFvs to the human PIM2 kinase [46]. Among the 12 *E. coli* clones that produced EcOX40-bound HuscFvs (by both indirect ELISA and HEK-OX40 cells), many clones were sibling clones; therefore, only seven clones (C2, D11, F8, C36, C41, C61, and C78) were studied further. 

Computerized simulation was used to predict the region of the EcOX40 protein that is bound by the HuscFvs to select the HuscFvs with OX40-stimulation potential. Satisfactory 3D structures of the HuscFvs of individual *E. coli* clones were built (i.e., the HuscFv residues in the most favored regions of the Ramachandran plots were higher than 90%) (Appendix A)) and used in the intermolecular docking with the full-length EcOX40 3D homology model [acquired by I-TASSER modeling using the crystal structure of PDB 2HEV [52] that was built from amino acids 29-214 of human OX40 (NP_003318.1)). It was found that HuscFvs of six clones (clones C2, D11, F8, C36, C41, and C78) interacted with the cysteine-rich domain-2 (CRD2) of EcOX40, which has been shown to function in OX40 stimulation [44], while the HuscFv of clone C62 formed a contact interface with CRD4 and the stalk of EcOX40, which do not participate in OX40 stimulation. Thus, HuscFvs of clones C2, D11, F8, C36, C41, and C78 were selected for further experiments.

The genes coding for HuscFvs (*huscfvs*) of clones C2, D11, F8, C36, C41, and C78 were subcloned from phagemids into protein expression plasmid vectors for large-scale production of the HuscFvs. High yields of HuscFvs were obtained from the sibling clones of C2, D11, C36, and C41 after subcloning. HuscFvC2, HuscFvD11, HuscFvC36, and HuscFvC41 were subjected to primary testing for their ability to stimulate OX40 signaling by using HEK-OX40 reporter cells prior to performing experiments that used freshly isolated human blood cells. For this experiment, the monomeric HuscFvs that bound to OX40 on the surface of HEK-OX40 reporter cells were crosslinked by an anti-His antibody. HuscFvC2 and HuscFvC41 could increase the luciferase activity of the reported cells indicating that they could stimulate downstream OX40 signaling to activate the NF-κB transcription factor. Therefore, HuscFvC2 and HuscFvC41 were used for preparing bivalent HuscFv-IgG1 Fc (Fcγ) fusion proteins, designated C2Fc and C41Fc fusion antibodies, respectively. Before using these further, the antigenic specificity of the fusion antibodies had to be verified; these were found to retain the EcOX40-binding capability of the original HuscFvs as tested by both co-immunoprecipitation with OX40 in the HEK-OX40 lysate and with EcOX40 on the HEK-OX40 cells, as well as on stimulated human T cells by flow cytometry. The EC50 values of the C2Fc and the C41Fc were also monitored, and they were in a nanomolar range (81.53 and 236.96 nM, respectively), which may require affinity improvement [75,76], if higher affinity is needed for in vivo use.

Naturally, naïve T cells require at least two signals to sustain T cell activation. The first activation signal is from the TCR-CD3 complex, and the second signal is from the interaction of the T cell CD28 with a B7 ligand (CD80) expressed on antigen-presenting cells. OX40 is expressed only on the simulated (effector) T cells. Thus, in our in vitro experiments for testing the T cell activating activities of the fusion antibodies, peripheral blood T cells isolated from the healthy donors had to be stimulated with anti-CD3 (first signal) and anti-CD28 (second signal) to induce OX40 expression on the T cell surface before the fusion antibodies were added to the cells; then, the expected outcomes of different experiments were observed. The prior stimulation by anti-CD3 and anti-CD28 (not the natural ligands) before adding the fusion antibodies is a limitation of our experiments, as the level of stimulation is not known and uncontrollable; if the cells were activated to the maximal or near-maximal capacity of their response, the additional stimulatory/expected effect after adding the fusion antibodies may not be detected. Fortunately, in most experiments, additional effects after adding the fusion antibodies to the CD3-CD28-stimulated T cells were observed, including an enhancement of T cell proliferation (Figure 7B–D), an ability to enhance cytokine production (Figure 8), an enhancement of T cell survival, and a reduction in total T cell death (Figure 9). Adding C41Fc to the CD3-CD28-activated total T cells and CD4^+^ T cells caused a statistically significant increment in granzyme B production; nevertheless, the small scales of granzyme B graphs (y axes of Figure 8D,H) suggest that the observed increases may not have a biological significance compared to the activated cells without the fusion antibodies. This may be due to the limitation of the in vitro condition of the experiment as stated above, or the amounts (and affinity) of the fusion antibodies may be suboptimal.

It is not possible to mimic the in vivo situation of cancer by the cell-based in vitro assay used in this study. Nevertheless, we successfully generated monocyte-derived DCs (MoDCs). The MoDCs were pulsed by an ovarian cancer-cell lysate and the pulsed DCs were used to stimulate the T cells of healthy donors. The addition of the fusion antibodies to tumor-antigen-primed T cells showed an enhancing trend in mediating the lysis of the cancer cells at an increasing T:E ratio. The relatively low enhancing activity of the fusion antibodies on the tumor-primed T cells might be due to their low target-binding affinity, which requires improvement (e.g., via CDR resurfacing); the number of the effector cells not being adequate (still too few); or because the MHCs of the effector cells (Thai donors) and the target cancer cells (Caucasian female with adenocarcinoma of the ovary) were only partially matched. Unfortunately, higher numbers of effector cells could not be tested due to the limitation in the whole blood volume from individual donors (ethical issue), and completely MHC-matched donors were not available. In real situations, the autologous target effector cell system exists, which might positively impact the capability of the fusion antibodies to enhance effector T cell activity. 

Between the two fusion antibodies, C2Fc seemed to perform better than C41Fc in promoting T cell proliferation and survival and reducing T cell death (Figure 7 and Figure 9), as well as enhancing cytokine (IL-2, TNF-α and IFN-γ) production (Figure 8). However, C41Fc seemed to perform better than C2Fc in inducing CD3-CD28-stimulated T cells (total T cell pool and CD4^+^ T cells) to produce more granzyme B compared with stimulated cells without the addition of fusion antibodies (Figure 8). Granzymes are serine proteases that play an important role in tumor cell killing by immune cells (CTL and NK). In addition to granzyme B, humans also have granzymes A, H, K, and M, the roles of which in the immune system are lesser known than that of granzyme B [76]. The role of granzyme-producing CD4^+^ T cells in antitumor immunity has emerged recently [76]. By using single-cell RNA-sequencing (RNA-seq) technology, it has become clear that several CD4^+^ T helper cell subtypes infiltrate into the tumor microenvironment (TME); these are cytolytic CD4^+^ T cells that express high levels of granzymes and perforin and may contribute to the anti-cancer response [77]. The novel multifaceted involvement of CD4^+^ T cells in the anti-tumor immune response has been reviewed along with their old paradigm in cancer [77]. Experiments using a fusion antibody made up of a hybrid HuscFvC2-HuscFvC41 linked to Fcγ to enhance T cell activities against tumors should be carried out.

In conclusion, novel OX40 agonistic antibodies that are fully human proteins in the form of a bivalent HuscFv-Fcγ have been generated. The fusion antibodies enhanced the proliferation, survival, and cytokine production of CD3-CD28-activated T cells. Under the limitation of the in vitro experiments, the fusion antibodies modestly enhanced granzyme B production from the activated T cells and showed trends in enhancing the anti-ovarian cancer activity of tumor-reactive T cells. Further step-by-step testing of fusion antibodies (after affinity improvement) will be a progression toward their use as an adjunctive agent in cancer immunotherapy.

## Figures and Tables

**Figure 1 vaccines-11-01826-f001:**
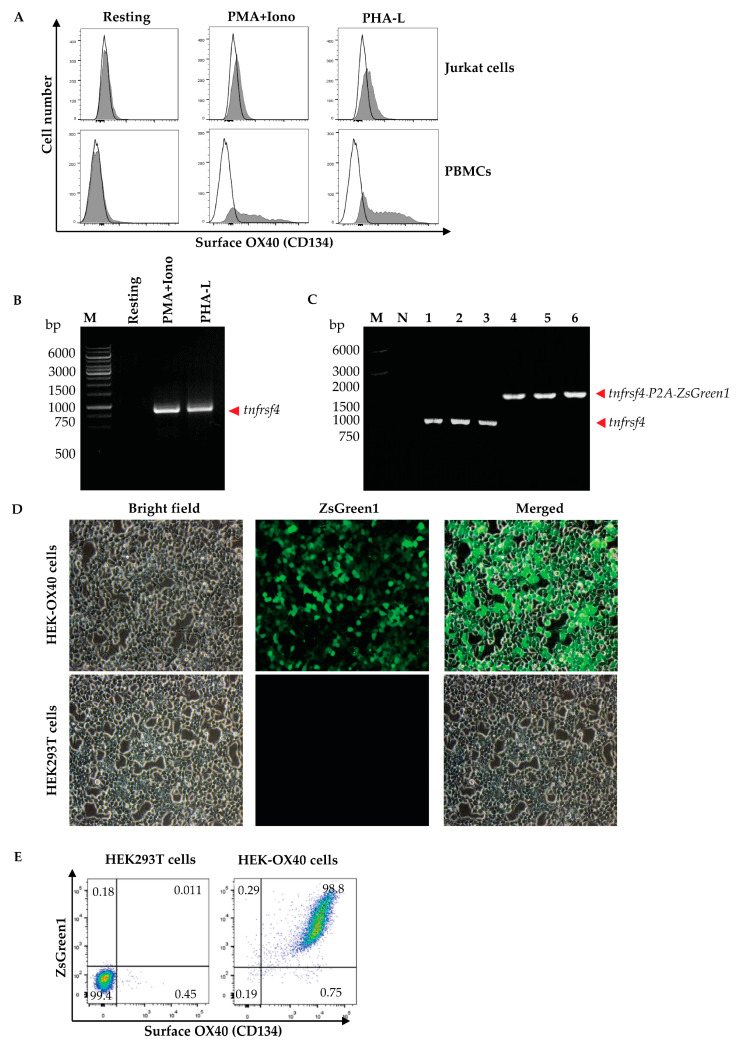
Native OX40 on the surface of activated Jurkat T cells and peripheral blood mononuclear cells (PBMCs) and the production of OX40-overexpressing cells (HEK-OX40 cells). (**A**) Expression of OX40 on the surface of Jurkat T cells and human PBMCs that were stimulated with 0.5 μg/mL of PMA and 1 μM Ionomycin (PMA + Iono) or 1 μg/mL PHA-L for 3 days. Resting cells without stimulation (Resting) served as controls. Activated and control cells were stained with anti-OX40-Alexa Fluor 647 and analyzed using flow cytometry. Gray histograms, OX40-expressed cells; white histograms, negative controls (cells without surface-exposed OX40). (**B**) Amplicons of *tnfrsf4* amplified from PMA + Iono- and PHA-L- activated PBMCs, respectively. Resting cells served as the control without *tnfrsf4* expression. Lane M, standard DNA ladder. (**C**) Amplicons of human *tnfrsf4* that were subcloned into the pJET1.2 cloning vector and transformed into *E. coli* (lanes 1–3) and amplicons of the bicistronic *P2A-ZsGreen1-*pLVX-puro vector from transformed *E. coli* clones (lanes 4–6). Lane M, 1 kb DNA ladder; lane N, negative control (no template). Numbers on the left of (**B**,**C**) are DNA sizes in base pairs (bp). (**D**) HEK293T cells infected with lentivirus containing *tnfrsf4-P2A-ZsGreen1* (upper blocks) compared with normal HEK293T cells (lower blocks). The ZsGreen1 protein was observed under a fluorescent microscope (green fluorescence). (**E**) Flow cytometry for determining the percentage of transformed HEK293T cells that expressed human OX40 (HEK-OX40 cells, 98.8%) compared with HEK293T cells.

**Figure 2 vaccines-11-01826-f002:**
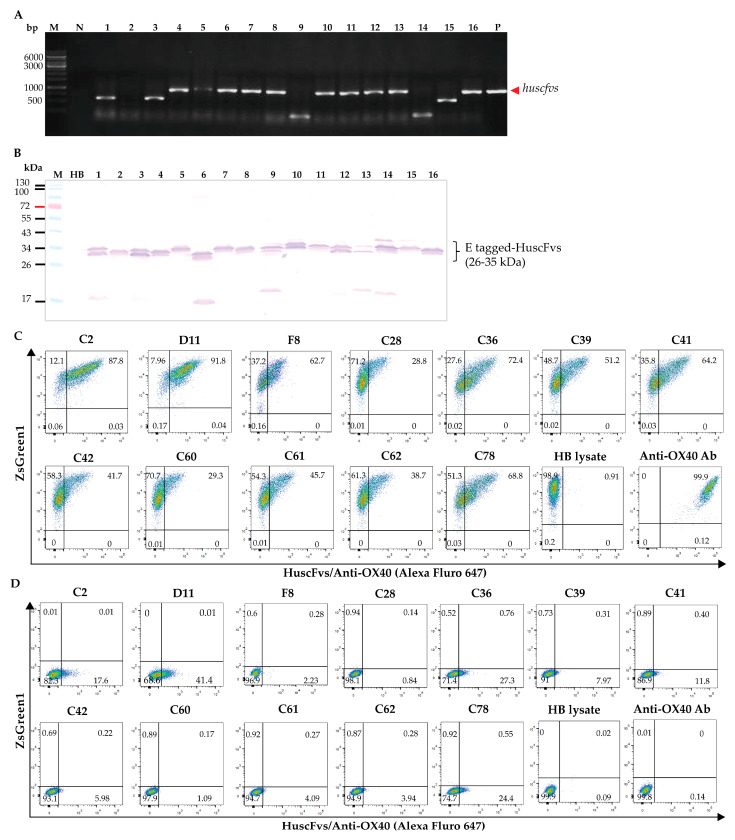
Phage panning and the generation of phage-transformed *E. coli* clones that produced EcOX40-bound HuscFvs. (**A**) Examples of *huscfv* amplicons from phage-infected HB2151 *E. coli* clones. Lane M, 1 kb DNA ladder; lane N, negative control (no DNA template); lane P, positive DNA amplicon control; lanes 4-8, 10-13 and 16, amplicons of *huscfvs* at ~1000 bp. (**B**) Western blot patterns of E-tagged-HuscFvs in lysates of representative *huscfv*-positive HB2151 *E. coli* clones (the expressed HuscFvs from some *E. coli* clones appeared as protein doublets at about 26–34 kDa; upper band, immature HuscFvs with a signal peptide; lower band, mature HuscFvs without a signal peptide). Lane M, protein standard marker; lanes 1–16, HuscFvs in lysates of *huscfv*-positive HB2151 *E. coli* clones. Numbers on the left are the molecular masses of the proteins in kDa. (**C**) HEK-OX40 cells incubated with HuscFvs of the 12 *E. coli* clones (C2, D11, F8, C28, C36, C39, C41, C42, C60, C61, C62, and C78) that showed positive binding to EcOX40 by the indirect ELISA. The lysate of the original HB2151 *E. coli* (HB) served as the no-HuscFv control and mouse anti-OX40-Alexa Fluor 647 served as the positive control. (**D**) HEK293 T cells were incubated with lysates of the 12 *E. coli* clones, HB, or mouse-anti-OX40-Alexa Fluor 647 as for (**C**). Thereafter, cells in both (**C**,**D**) were probed with the rabbit anti-E-tag antibody and goat anti-rabbit IgG-Alexa Fluor 647 before being subjected to flow cytometry.

**Figure 3 vaccines-11-01826-f003:**
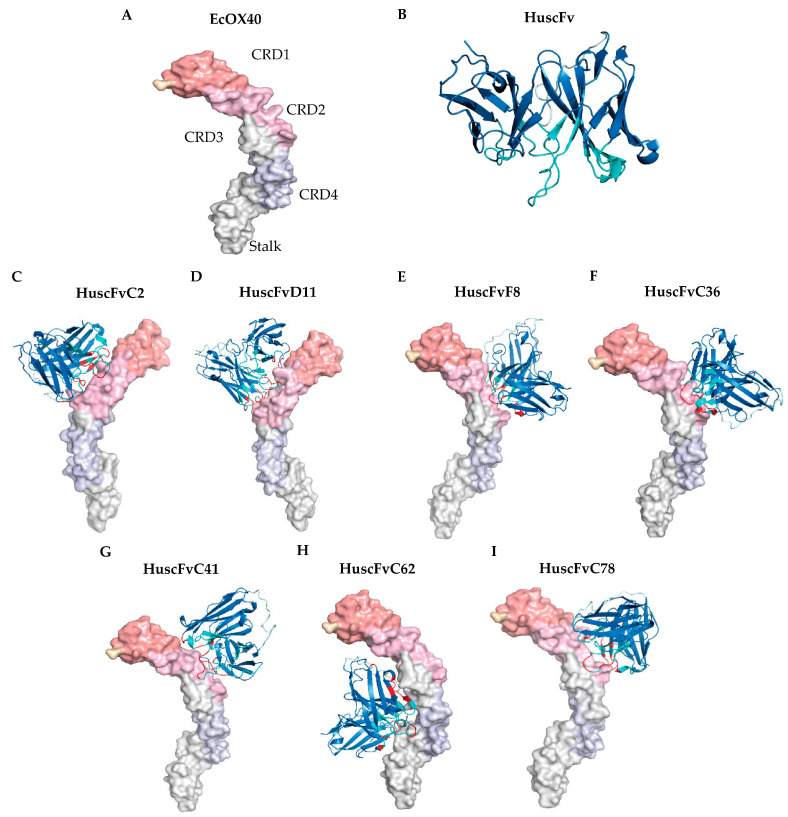
Computerized simulation for predicting the regions of EcOX40 that formed contact interfaces with the HuscFvs. (**A**) Three-dimensional structure of a modeled monomeric EcOX40 showing four cysteine-rich domains (CRD1-CDR4) and the stalk. (**B**) Three-dimensional structure of the modeled HuscFv. (**C**–**I**) Intermolecular docking of HuscFvs from the seven *E. coli* clones and EcOX40.

**Figure 4 vaccines-11-01826-f004:**
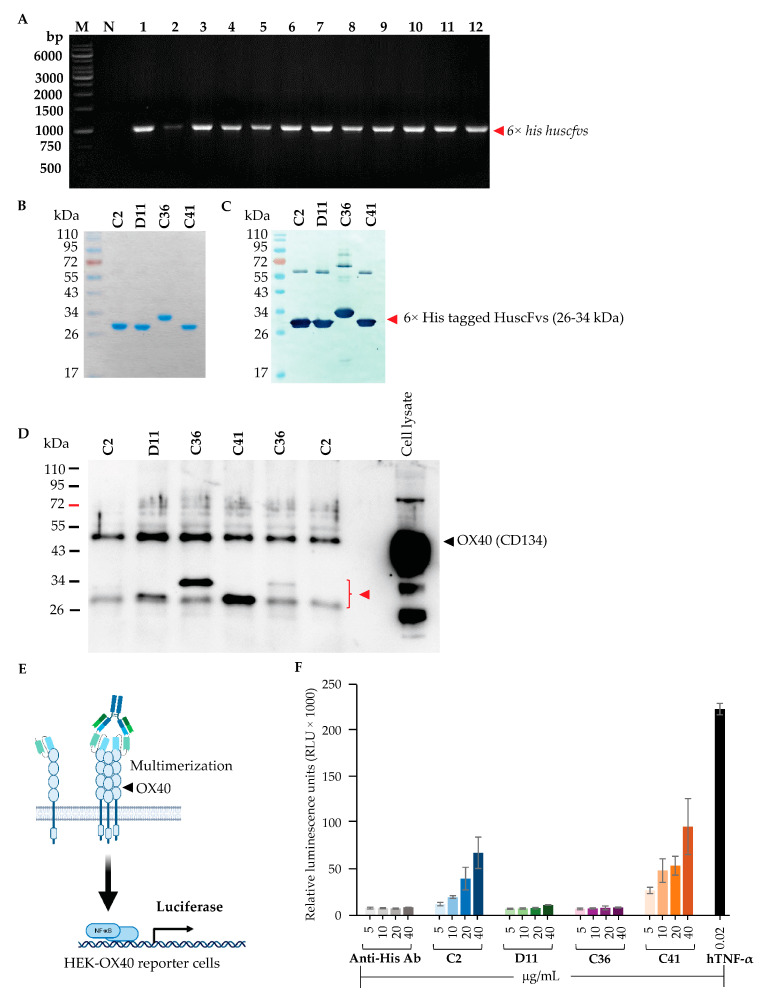
Production of 6× His-tagged HuscFvs and the NF-κB-activating activity of the HuscFvs to EcOX40. (**A**) Amplicons of the DNA coding for 6× His-tagged HuscFvs (red arrowhead) that were amplified from recombinant-pET23b+-transformed BL21 (DE3) *E. coli* by direct colony PCR. Lane M, 1 kb DNA ladder; lane N, negative control (no template); lanes 1–3, 4–6, 7–9, and 10–12 are amplicons of *huscfv*C2, *huscfv*D11, *huscfv*C36, and *huscfv*C41, respectively. (**B**) SDS-PAGE and CBB-stained patterns of purified HuscFvC2, HuscFvD11, HuscFvC36, and HuscFvC41 (between 26–34 kDa). (**C**) Western blot patterns of purified 6× His-tagged HuscFvC2, HuscFvD11, HuscFvC36, and HuscFvC41 (red arrowhead). (**D**) HuscFvs (red arrowhead) were coprecipitated with OX40 from the lysate of HEK-OX40 cells. In this experiment, purified 6× His-tagged HuscFvs were incubated with the HEK-OX40 cell lysate, and the HuscFv-OX40 complexes were pulled down by mouse anti-His-coated protein G magnetic beads. The co-immunoprecipitated proteins were eluted from the beads and subjected to Western blot analysis. In the Western blot analysis, bands of SDS-PAGE-separated OX40 derived from the HEK-OX40 cells were probed with a rabbit anti-OX40 antibody whereas the HuscFvs were detected using a mouse anti-His antibody. Bands representing OX40 (black arrowhead, ~50 kDa) and HuscFvs (red arrowhead, 26–34 kDa) were both detected in the same eluate from the anti-His-coated protein G magnetic beads. OX40 in the lysate of HEK-OX40 cells (“Cell lysate” in the figure) served as the OX40 marker. Numbers at the left of (**B**–**D**) are the protein masses in kDa. (**E**) Schematic diagram of the assay for determining effect of crosslinked (multimeric) HuscFvs on the OX40-mediated activation of NF-κB transcription factor activity (increased luciferase activity in HEK-OX40 reporter cells). (**F**) HEK-OX40 reporter cells were treated with various concentrations of HuscFvs (5–40 μg/mL), and the HuscFvs were crosslinked with a mouse anti-His antibody. The luciferase activities were measured after 6 h of incubation. Human TNF-α (hTNF-α) was used as a positive NF-κB-activation control; an anti-His antibody (Anti-His Ab) was used as the negative activation control. Crosslinked HuscFvC2 and HuscFvC41 could stimulate HEK-OX40 reporter cells, causing an increment in luciferase expression in a dose-dependent manner.

**Figure 5 vaccines-11-01826-f005:**
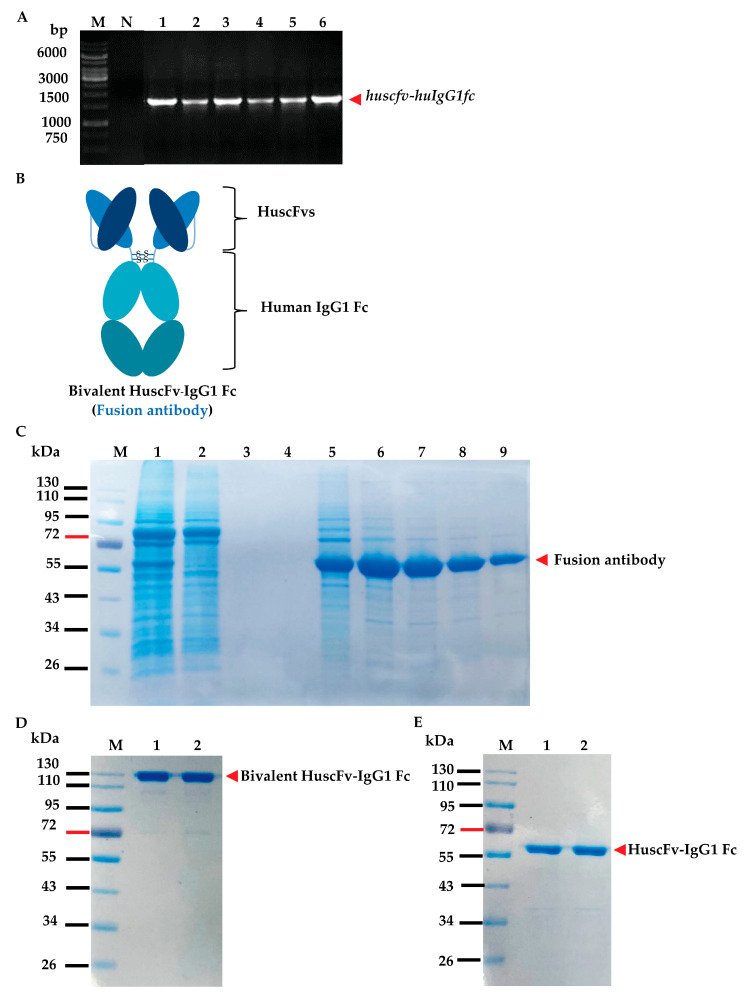
Production of bivalent HuscFv-Fcγ fusion proteins (fusion antibodies). (**A**) Amplicons of *huscfvs-huIgG1 fc* amplified from recombinant pINFUSE-Fc2 transformed JM109 *E. coli* clones by direct colony PCR (arrowhead). Lane M, 1 kb DNA ladder; N, negative control (no template); lanes 1–3, *huscfvC2-huIgG1 fc*; lanes 4–6, *huscfvC41-huIgG1 fc* amplicons. (**B**) Diagram of the bivalent HuscFv-huIgG1 Fc (fusion antibody). (**C**) SDS-PAGE and CBB-stained fusion antibodies (bivalent HuscFvC2-IgG1 Fc) in various purification fractions. Fusion antibodies were purified from the culture supernatant of *huscfv-huIgG1 Fc2* pINFUSE-transfected HEK293E suspension cells by protein G affinity column chromatography. Lane M, protein standard markers; lane 1, transfected HEK293E cell culture supernatant; lane 2, flow through; lanes 3–9, eluted fractions containing the fusion antibodies. (**D**,**E**) Fusion antibodies were subjected to SDS-PAGE under non-reducing conditions and reducing conditions, respectively. Non-reduced fusion antibodies appear as bands at ~110 kDa (arrowhead, bivalent HuscFvs-IgG1 Fc); reduced fusion antibodies appear as ~55 kDa bands (arrowhead, HuscFv-IgG1 Fc). Lane M, protein standard markers; lane 1, monovalent HuscFvC2 fusion antibody; lane 2, monovalent HuscFvC41 fusion antibody. Numbers on the left of (**C**–**E**) are the molecular masses of proteins in kDa.

**Figure 6 vaccines-11-01826-f006:**
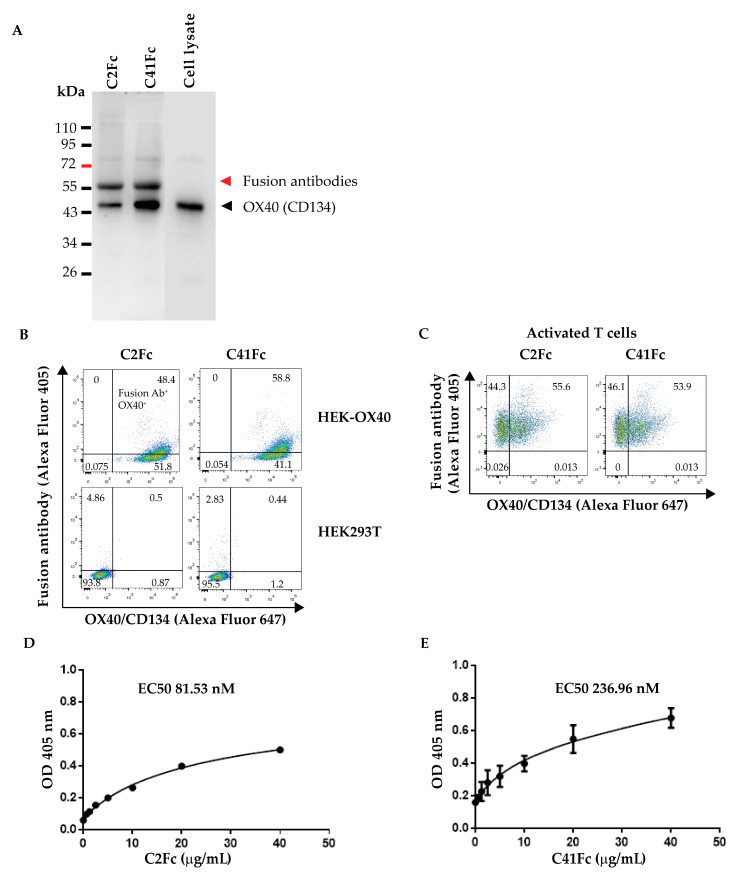
Binding of bivalent HuscFv−Fcγ fusion proteins (fusion antibodies) to OX40. (**A**) Co−immunoprecipitation of fusion antibodies with OX40 in the HEK−OX40 cell lysate. Fusion antibodies C2Fc and C41Fc were incubated with the HEK−OX40 cell lysate. The complexes were pulled down by protein G magnetic beads, eluted from the beads, and subjected to Western blot analysis. The Western blot pattern of OX40 in the lysate of HEK−OX40 alone was used for comparison. The OX40 protein bands were detected by using a rabbit anti-OX40 antibody and the fusion antibodies were detected using a goat anti−huIgG (H + L) −HRP conjugate. The immunoprecipitated complexes revealed two proteins, i.e., OX40 bands at ~50 kDa while bands of the fusion antibodies appeared at ~55 kDa (under reduced condition of the SDS−PAGE), indicating that the fusion antibodies could bind to the OX40 in the HEK−OX40 lysate, and that both were coprecipitated. Numbers at the left of (**A**) are the protein masses in kDa. (**B**) Binding of fusion antibodies to EcOX40 on the HEK−OX40 cell surface. HEK−OX40 cells were added with 10 μg/mL of fusion antibodies and counter−stained with an anti−huIgG Fc−Alexa Fluor 405 conjugate. The stained cells were assessed by flow cytometry. (**C**) Human T cells were stimulated with PHA−L for 3 days to induce expression of OX40. Fusion antibodies, the anti−huIgG Fc−Alexa Fluor 405 conjugate, and anti−OX40−Alexa Fluor 647 were then added to the cells. The stained cells were assessed by flow cytometry. (**D**,**E**) Half−maximal effective concentration (EC50) of the two fusion antibodies to EcOX40. Fusion antibodies at 0.3125–40 μg/mL were tested for their binding affinity to EcOX40 by indirect ELISA; the calculated EC50 of the C2Fc and C41Fc were 81.53 and 236.96 nM, respectively.

**Figure 7 vaccines-11-01826-f007:**
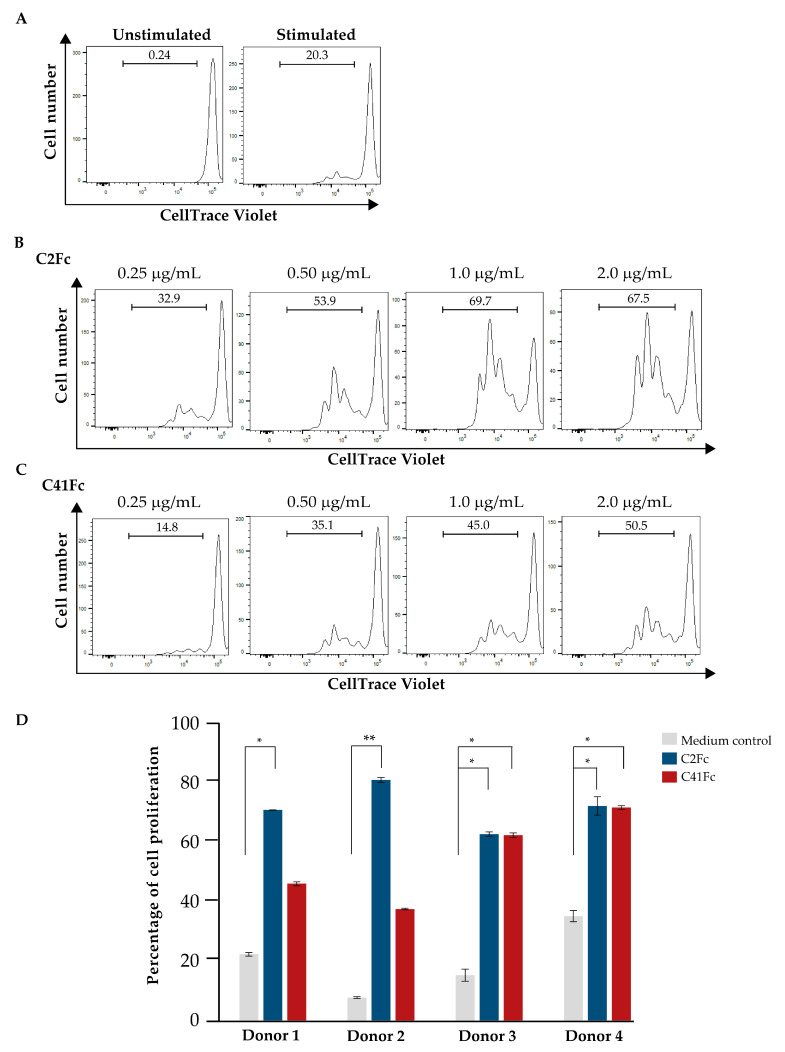
Fusion antibodies to EcOX40 enhanced the proliferation of stimulated T cells. Tissue culture plates were precoated with 0.5 μg/mL of anti-CD3 antibody and different concentrations of fusion antibodies (0.25, 0.50, 1.0, and 2.0 μg/mL). T cells isolated from human peripheral blood were labeled with CellTrace Violet. The labeled T cells and a soluble anti-CD28 antibody (1 μg/mL) were added into each precoated well. The cells were cultured for 5 days and harvested to measure the reduction of CellTrace violet by flow cytometry. (**A**) Histograms showing unstimulated T cells (left) and CD3-CD28-stimulated T cells (right). (**B**) Histograms showing the enhanced proliferation of CD3-CD28-stimulated T cells by various concentrations of immobilized C2Fc. (**C**) Histograms showing enhanced the proliferation of CD3-CD28-stimulated T cells by various concentrations of immobilized C41Fc. (**D**) Percentages of proliferating CD3-CD28-stimulated T cells after exposure to 1.0 μg/mL (from titration in (**B**,**C**)) of C2Fc (blue bars) and C41Fc (orange bars) for 5 days compared with CD3-CD28-stimulated T cells in medium alone (gray bars). Experiments were performed with T cells isolated from four healthy donors (donors 1–4). Each treatment was performed in triplicate. Four independent and reproducible experiments were performed, and the data were combined and averaged. The data are expressed as the means ± SD; difference is statistically significant at * *p* < 0.05 and ** *p* < 0.01.

**Figure 8 vaccines-11-01826-f008:**
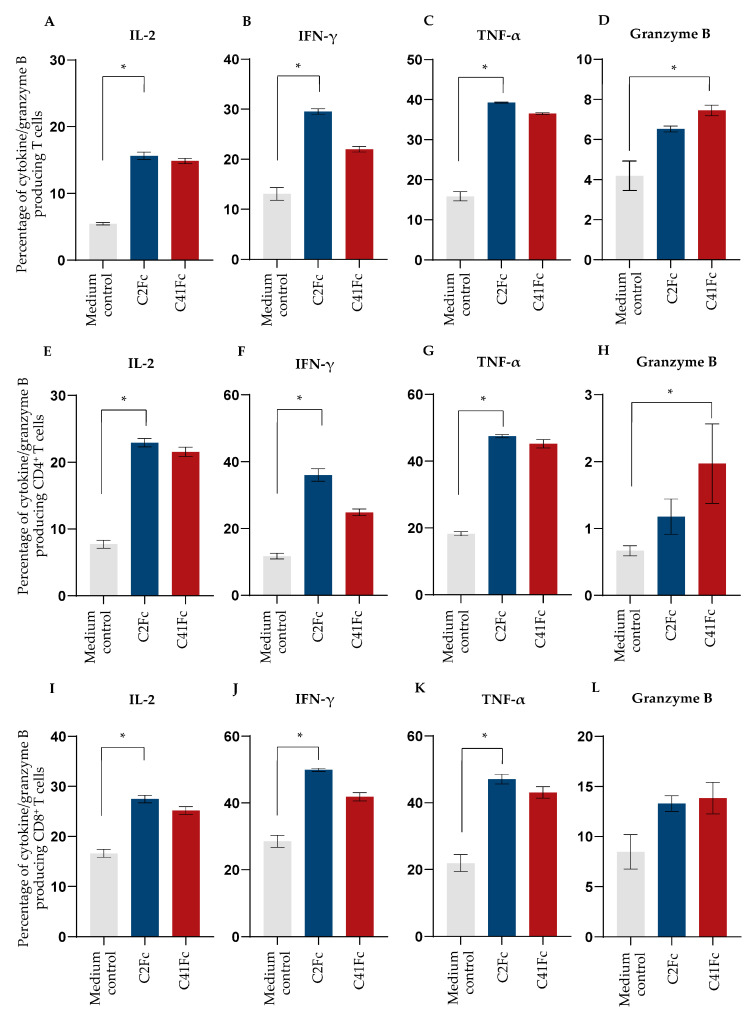
Fusion antibodies to EcOX40 caused an increment in the percentages of cytokine/granzyme B-producing T cells. Human T cells were stimulated with 1 μg/mL of immobilized anti-CD3 and 1 μg/mL of soluble anti-CD28 antibody with/without 1 μg/mL of co-immobilized C2Fc/C41Fc for 2 days. Then, Brefeldin A, a cytokine-blocking solution, was added into each well and incubated for 16 h. The cells were then harvested and fixed with 4% paraformaldehyde-PBS. Surface CD3, CD4, and CD8 were then stained with their respective antibodies, and the intracellular cytokines and granzyme B were then stained with anti-IL-2, anti-IFN-γ, anti-TNF-α, or anti-granzyme B antibodies, respectively. The stained cells were analyzed using flow cytometry. (**A**–**D**) Percentages of IL-2, IFN-γ, TNF-α, and granzyme B-producing cells, respectively, in the CD3^+^ T cell population. (**E–H**) Percentages of IL-2, IFN-γ, TNF-α, and granzyme B-producing cells, respectively, in the CD4^+^ T cell population. (**I–L**) Percentages of IL-2, IFN-γ, TNF-α, and granzyme B-producing cells, respectively, in the CD8^+^ T cell population. Gray bars, control T cells, i.e., CD3-CD28-stimulated T cell populations without fusion antibodies (medium alone). Blue and orange bars, CD3-CD28-stimulated T cell populations treated with C2Fc and C41Fc fusion antibodies, respectively. The data are expressed as the means ± SD; difference is statistically significant at * *p* < 0.05.

**Figure 9 vaccines-11-01826-f009:**
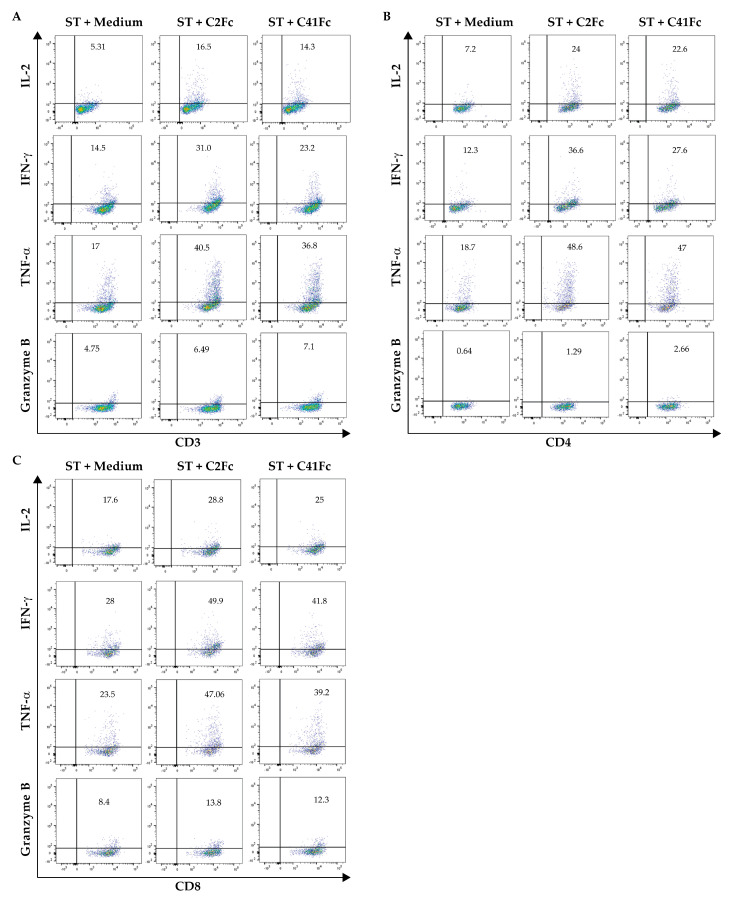
The representative gates of flow cytometric dot plots showing percentages of cytokine/granzyme B−producing cells. (**A**) CD3−CD28−activated CD3^+^ T cells, (**B**) CD3−CD28−activated CD3^+^ CD4^+^ T cells, and (**C**) CD3−CD28−activated CD3^+^ CD8^+^ T cells exposed to fusion antibodies compared to activated cells in medium alone.

**Figure 10 vaccines-11-01826-f010:**
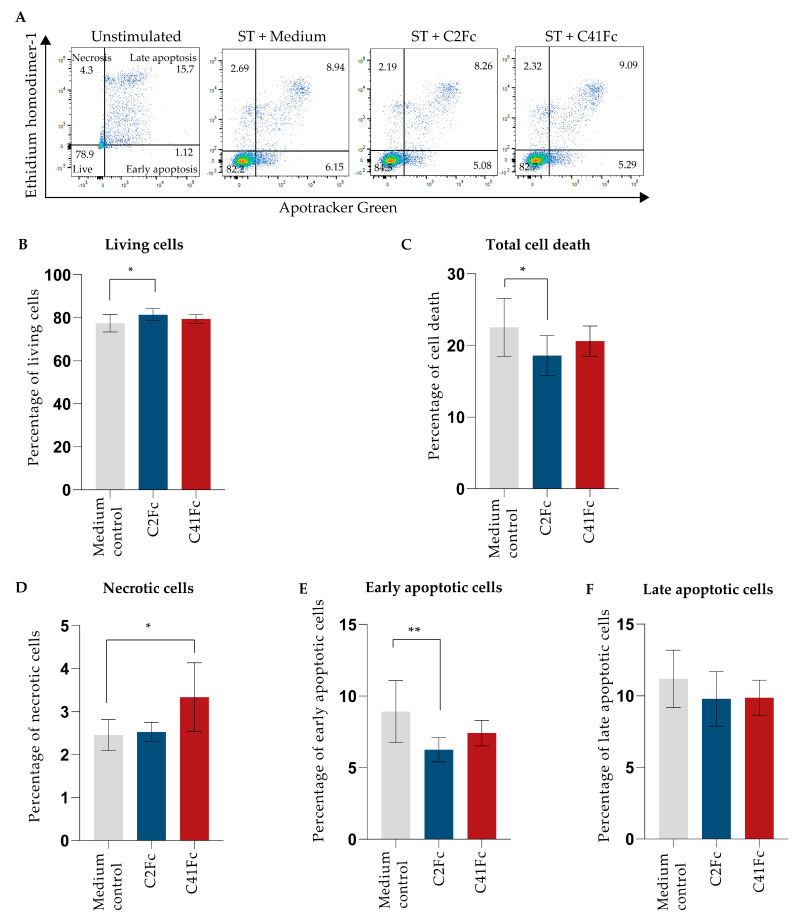
Fusion antibodies to OX40 increased T cell survival and reduced T cell death. Human T cells were stimulated with 1 μg/mL of immobilized anti−CD3 and 1 μg/mL of soluble anti−CD28 antibody with/without 1 μg/mL of co−immobilized C2Fc or C41Fc for 5 days. The cells were then harvested and stained with Apotracker Green and ethidium homodimer 1 (EthD−1). Cell death and apoptosis were analyzed using flow cytometry. (**A**) Dot plots representing living cells (Apotracker green**-**, EthD−1**-**), necrotic cells (Apotracker green**-**, EthD−1**+**), early apoptotic cells (Apotracker green**+**, EthD−1**-**), and late apoptotic cells (Apotracker green**+**, EthD−1**+**). (**B**) Percentages of living cells. (**C**) Percentages of total cell death (necrosis and late apoptosis). (**D**) Percentages of necrotic cells. (**E**,**F**) Percentages of early and late apoptotic cells, respectively. Gray bars, CD3−CD28−stimulated T cells in medium alone. Blue and orange bars, CD3−CD28−stimulated T cells added with C2Fc and C41Fc fusion antibodies added, respectively. ST, CD3−CD28−stimulated T cells. Experiments were performed with T cells isolated from three healthy donors (donors 1–3); each treatment in an experiment was performed in triplicate. The data are expressed as the means ± SD; difference is statistically significant at * *p* < 0.05 and ** *p* < 0.01.

**Figure 11 vaccines-11-01826-f011:**
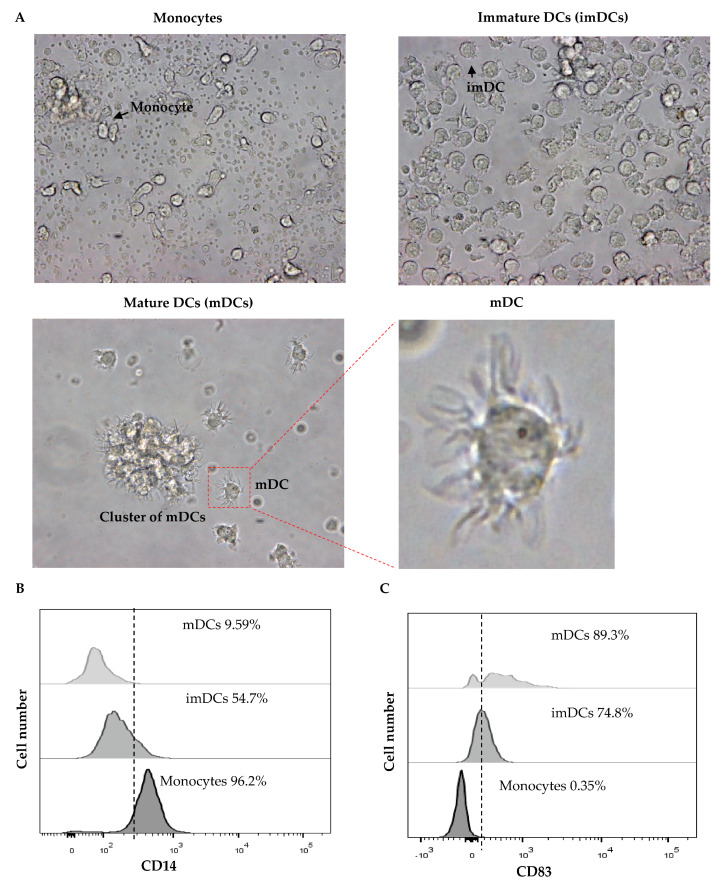
Generation of monocyte-derived dendritic cells (MoDCs). Monocytes were isolated from PBMCs and cultured in 10% FBS−AIM−V supplemented with GM−CSF and IL−4 for 5 days to generate immature DCs (imDCs). Immature DCs were then differentiated further into mature DCs by adding TNF−α and CD40L to the cell culture for 2 days. (**A**) Microscopic appearances of monocytes, imDCs and mDCs. (**B**) Results of flow cytometry for determining the percentages of CD14−expressed cells during the process of preparing monocyte−derived DCs. CD14 (marker of monocytes) was gradually downregulated as the monocytes differentiated into imDCs and mDCs. (**C**) Results of flow cytometry for determining the percentages of cells that expressed CD83 (marker of mature DCs) during the differentiation of monocytes into imDCs and mDCs.

**Figure 12 vaccines-11-01826-f012:**
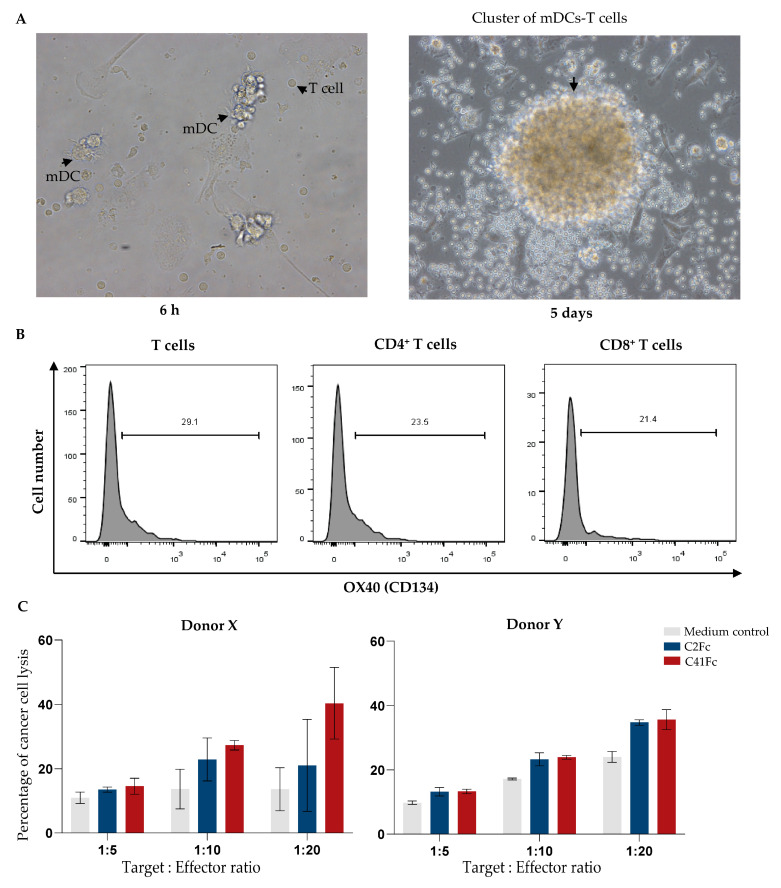
Generation of tumor-antigen-primed T cells (tumor-reactive T cells) and cytotoxic activities of the effector T cells to ovarian cancer cells in the presence/absence of fusion antibodies. (**A**) Mature DCs were pulsed with 100 μg/mL of SK-OV-3 lysate for 12 h. Autologous T cells were cocultured with the pulsed-DCs at a ratio of DCs:T cells of 1:5. After 5 days of coculturing, tumor-lysate-pulsed mDCs induced the T cells to proliferate and to form large cell clusters. The cells from the coculture were stained for surface CD3, CD4, CD8, and OX40 and analyzed via flow cytometry. (**B**) Histograms showing OX40 expression on CD3^+^ T cells, CD4^+^ T cells, and CD8^+^ T cells. (**C**) Cytotoxic activity of T cells determined by using the calcein-release assay. Tumor-antigen-primed T cells were pre-activated with C2Fc, C41Fc, or medium alone for 24 h. Then, they were co-cultured with calcein-AM-labeled SK-OV-3 cells (MHC-class I partially matched) at a target/effector ratio (T:E) of 1:5, 1:10, or 1:20. After 24 h, culture supernatants were collected and the calcein released from the SK-OV-3 cells was measured. The percentages of specific lysis are shown as bar graphs. Gray bars, medium controls; blue and orange bars, C2Fc and C41Fc fusion antibodies treated cells, respectively. Results from two replicated experiments from the two blood donors are shown.

**Figure 13 vaccines-11-01826-f013:**
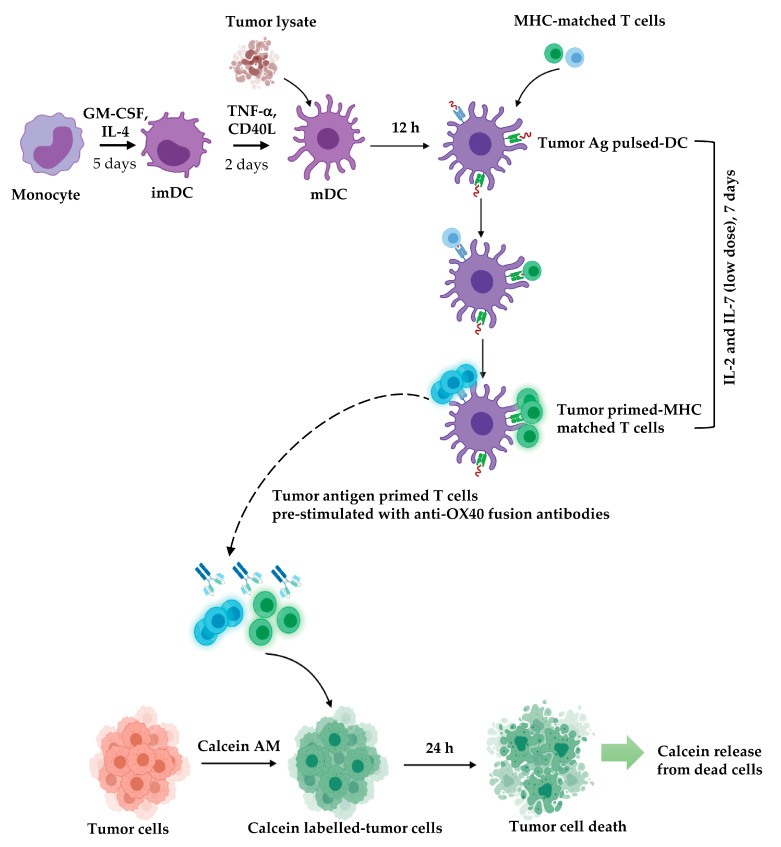
Experimental steps for testing the ability of fusion antibodies to enhance the anti-cancer activity of tumor-antigen-primed T cells. The figure was created using BioRender.com (accessed on 11 August 2021).

## Data Availability

The dataset generated during the present study is accessible from the corresponding author upon reasonable request. Requests for materials can be directed to wanpen.cha@mahidol.ac.th. All materials and reagents will be made available upon the installment of a material transfer agreement (MTA).

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
