# Peer review of "Agonistic Bivalent Human scFvs-Fcγ Fusion Antibodies to OX40 Ectodomain Enhance T Cell Activities against Cancer"

_vaccines, 2023, doi:10.3390/vaccines11121826_

Round 1

Reviewer 1 Report

Comments and Suggestions for Authors

In the proposed manuscript, Mahasongkram and colleagues present the production and characterization of novel human bivalent agonistic fusion antibodies to OX40/CD134. Concerning the production phase, the use of techniques as phage display and genetic engineering are well explained and robust. However, despite an initial characterization of the antibodies was performed, their functionality is not completely convincing and some issues are raised in the proposed revision.

In general, the topic is relevant but not really novel, as exposed in the Introduction (lines 103-117). The open questions, mainly concerning the effect of the antibodies on Tregs and on signalling cascade, actually are not fully addressed and solved by the data.

Major points

Fig. 1.

The use of Jurkat cells is not explained, nor detailed in Methods section.

Fig 7C.

The selected concentration 1ug/mL of the fusion antibodies seems suboptimal for C41Fc, that may explain its lower activity compared to C2Fc shown in the following figures. What is the reason for choosing the same condition in the two cases, given the different biological outcome?

Fig 8.

Even if statistically different, y axis scales of Granzyme B graphs suggest that the observed increases may not have a biological meaning (eg from 4 to 7 % of producing T cells). Authors should discuss this point.

Differences of basal level values (“medium control”, gray bars) among the stainings of the same T cell group should be justified (eg from 0.5 to 20% approximately in CD4 T cells group).

Representative gates/histograms should be shown, as in Fig 9A.

Fig 9B.

Bars do not seem to be significantly different, in spite of statistics.

Fig 11C.

Data are not completely convincing. Authors state that “tumor antigen primed-T cells exposed to C2Fc/C41Fc fusion antibodies had a trend of increasing ability in tumor cell killing when the effector cell numbers were increased, compared to tumor antigen primed-T cells in medium control (no fusion antibodies)”. This is true for C41Fc in Donor X, while standard deviations of C2Fc are too high to draw any conclusion. Concerning Donor Y, the increasing trend of cancer cell lysis caused by the fusion antibodies is comparable to the one in medium control situation. Authors should discuss and / or mitigate their sentence, also in the dedicated part of the Discussion (lines 1083-1084).

Minor points

Fetal bovine serum is used to culture PBMCs instead of human one: what is the reason why?

“gramzyme” in Fig 8.

Sentence in lines 27-28 is not clear.

Sentence in lines 57-58 is not clear.

Line 86: binds.

Sentence in lines 100-102 is not clear.

Line 990: co-culturing.

In general, higher quality images should be provided.

Comments on the Quality of English Language

Few minor revision on English Language are needed (see Minor Points of the revision).

Author Response

Reviewer 1

In the proposed manuscript, Mahasongkram and colleagues present the production and characterization of novel human bivalent agonistic fusion antibodies to OX40/CD134. Concerning the production phase, the use of techniques such as phage display and genetic engineering are well explained and robust. However, despite an initial characterization of the antibodies being performed, their functionality is not completely convincing, and some issues are raised in the proposed revision.

In general, the topic is relevant but not novel, as exposed in the Introduction (lines 103-117). The open questions, mainly concerning the effect of the antibodies on Tregs and on signaling cascade, are not fully addressed, and solved by the data.

Major points

Comment:

Fig. 1.

The use of Jurkat cells is not explained, nor detailed in Methods section.

Response:          

We have added the details of Jurkat cells to “Methods” sections 2.2 and 2.4 of the revised manuscript. Thank you very much.

Comment:

Fig 7C.

The selected concentration 1 mg/mL of the fusion antibodies seems suboptimal for C41Fc that may explain its lower activity compared to C2Fc shown in the following figures. What is the reason for choosing the same condition in the two cases, given the different biological outcome?

Response:

We chose this concentration of fusion antibodies because this concentration gave reproductive results and could show the difference between medium control and fusion antibodies. It might be suboptimal as per your comment. We have added your point to the text of the revised manuscript (Lines 890-893 and 1193-1199). Thank you very much.

Comment:

Fig 8.

Even if statistically different, y axis scales of Granzyme B graphs suggest that the observed increases may not have a biological meaning (e.g., from 4 to 7 % of producing T cells). Authors should discuss this point.

Response:

Thank you very much. We have added your point to “Discussion”, i.e., Lines 1193-1199 of the revised manuscript.

Comment:

Differences of basal level values (“medium control”, gray bars) among the staining of the same T cell group should be justified (e.g., from 0.5 to 20% approximately in CD4 T cells group).

Representative gates/histograms should be shown, as in Fig 9A.

Response:

In Figure 8, the basal levels of cytokines in the medium controls were different because different types of cytokines (IL-2, IFN-γ, TNF-α, and granzyme B) can be produced at different levels by different subsets of T cells. We show the percentages of cytokine producing cells in each subset of T cells in in vitro stimulation and the ability of the fusion antibodies in enhancement of cytokine production. The normalization data are presented in fold changes of cytokines compared to respective medium controls.

The representative gates of flow cytometric dot plots are added to new Figure 9 of the revised manuscript. Thus, Figures 9, 10, 11 and 12 of the original manuscript were changed to Figures 10, 11, 12 and 13, respectively, of the revised manuscript.

Thank you very much.

Comment:

Fig 9B.

Bars do not seem to be significantly different, in spite of statistics.

Response:

In this study, we use ANOVA-Kruskal Wallis test. We found that medium control and C2Fc are significantly different (p-value at 0.0306) as shown in the table below.

Live

Dunn's multiple comparisons test

Mean rank diff.

Significant?

Summary

Adjusted p value

Medium vs C2Fc

-11.04

Yes

*

0.0306

Medium vs C41Fc

-3.333

No

ns

> 0.9999

C2Fc vs C41Fc

7.708

No

ns

0.2186

Comment:

Fig 11C.

Data are not completely convincing. Authors state that “tumor antigen primed-T cells exposed to C2Fc/C41Fc fusion antibodies had a trend of increasing ability in tumor cell killing when the effector cell numbers were increased, compared to tumor antigen primed-T cells in medium control (no fusion antibodies)”. This is true for C41Fc in Donor X, while standard deviations of C2Fc are too high to draw any conclusion. Concerning Donor Y, the increasing trend of cancer cell lysis caused by the fusion antibodies is comparable to the one in medium control situation. Authors should discuss and / or mitigate their sentence, also in the dedicated part of the Discussion (lines 1083-1084).

Response:

We have rephrased the sentences in the Discussion (Lines 1193-1199), Conclusion (Lines 1131-1138) and Abstract of the revised manuscript as per your suggestion. Thank you very much.

Minor points

Comment:

Fetal bovine serum is used to culture PBMCs instead of human one: what is the reason why?

Response:

Thank you very much. Your point is very well taken. However, we used fetal bovine serum (FBS) because it is commonly used as a supplement for cell cultures for providing the necessary nutrients for the growth of most cell lines and primary cells. We must confess that we did not apply to the IRB for the use of human serum.

Comment:

“gramzyme” in Fig 8.

Response:

The word “gramzyme” in y axes of Figure 8 was changed to “granzyme. Thank you very much.  

Comment:

Sentence in lines 27-28 is not clear.

Response:

Binding of OX40 ligand (OX40L) at the cysteine-rich domain 2 of extracellular portion of OX40 on T cells has been shown previously to cause downstream signaling in T cells leading to activation of NF-kB transcription factor and gene expression.

Comment:

Sentence in lines 57-58 is not clear.

Response:

We mean immune surveillance by host natural killer cells which is anti-cancer innate immunity at homeostasis

Comment:

Line 86: binds.

Response:

The word “binds” was corrected to “bind” (Line 87 of the revised manuscript). Thank you.

Comment:

Sentence in lines 100-102 is not clear.

Response:

We have rephrased the sentence (Lines 101-102 of the revised manuscript). Thank you.

Comment:

Line 990: co-culturing.

Response:

The word “co-culturing” in Line 990 was changed to co-cultured” in Line 1013-1014 of the revised manuscript. Thank you.

Comment:

In general, higher quality images should be provided.

Responses:

We have changed Figures 1E, 2C, 2D, 6B, 6C, 7A-7C and 10A to improve the quality of the images. All Figures are 700 dpi (as per the journal requirement). Thank you.

Comments on the Quality of English Language

Few minor revisions on English Language are needed (see Minor Points of the revision).

Response:

Thank you very much. We have revised them accordingly.

On behalf of all authors, I would like to thank you for your time spent on reading our manuscript and your leaned comments and suggestions. Our appreciations.

                                                                                 Yours sincerely,

                                                                  Wanpen Chaicumpa, D.V.M. (Hons.), Ph. D.

Reviewer 1

In the proposed manuscript, Mahasongkram and colleagues present the production and characterization of novel human bivalent agonistic fusion antibodies to OX40/CD134. Concerning the production phase, the use of techniques such as phage display and genetic engineering are well explained and robust. However, despite an initial characterization of the antibodies being performed, their functionality is not completely convincing, and some issues are raised in the proposed revision.

In general, the topic is relevant but not novel, as exposed in the Introduction (lines 103-117). The open questions, mainly concerning the effect of the antibodies on Tregs and on signaling cascade, are not fully addressed, and solved by the data.

Major points

Comment:

Fig. 1.

The use of Jurkat cells is not explained, nor detailed in Methods section.

Response:          

We have added the details of Jurkat cells to “Methods” sections 2.2 and 2.4 of the revised manuscript. Thank you very much.

Comment:

Fig 7C.

The selected concentration 1 mg/mL of the fusion antibodies seems suboptimal for C41Fc that may explain its lower activity compared to C2Fc shown in the following figures. What is the reason for choosing the same condition in the two cases, given the different biological outcome?

Response:

We chose this concentration of fusion antibodies because this concentration gave reproductive results and could show the difference between medium control and fusion antibodies. It might be suboptimal as per your comment. We have added your point to the text of the revised manuscript (Lines 890-893 and 1193-1199). Thank you very much.

Comment:

Fig 8.

Even if statistically different, y axis scales of Granzyme B graphs suggest that the observed increases may not have a biological meaning (e.g., from 4 to 7 % of producing T cells). Authors should discuss this point.

Response:

Thank you very much. We have added your point to “Discussion”, i.e., Lines 1193-1199 of the revised manuscript.

Comment:

Differences of basal level values (“medium control”, gray bars) among the staining of the same T cell group should be justified (e.g., from 0.5 to 20% approximately in CD4 T cells group).

Representative gates/histograms should be shown, as in Fig 9A.

Response:

In Figure 8, the basal levels of cytokines in the medium controls were different because different types of cytokines (IL-2, IFN-γ, TNF-α, and granzyme B) can be produced at different levels by different subsets of T cells. We show the percentages of cytokine producing cells in each subset of T cells in in vitro stimulation and the ability of the fusion antibodies in enhancement of cytokine production. The normalization data are presented in fold changes of cytokines compared to respective medium controls.

The representative gates of flow cytometric dot plots are added to new Figure 9 of the revised manuscript. Thus, Figures 9, 10, 11 and 12 of the original manuscript were changed to Figures 10, 11, 12 and 13, respectively, of the revised manuscript.

Thank you very much.

Comment:

Fig 9B.

Bars do not seem to be significantly different, in spite of statistics.

Response:

In this study, we use ANOVA-Kruskal Wallis test. We found that medium control and C2Fc are significantly different (p-value at 0.0306) as shown in the table below.

Live

Dunn's multiple comparisons test

Mean rank diff.

Significant?

Summary

Adjusted p value

Medium vs C2Fc

-11.04

Yes

*

0.0306

Medium vs C41Fc

-3.333

No

ns

> 0.9999

C2Fc vs C41Fc

7.708

No

ns

0.2186

Comment:

Fig 11C.

Data are not completely convincing. Authors state that “tumor antigen primed-T cells exposed to C2Fc/C41Fc fusion antibodies had a trend of increasing ability in tumor cell killing when the effector cell numbers were increased, compared to tumor antigen primed-T cells in medium control (no fusion antibodies)”. This is true for C41Fc in Donor X, while standard deviations of C2Fc are too high to draw any conclusion. Concerning Donor Y, the increasing trend of cancer cell lysis caused by the fusion antibodies is comparable to the one in medium control situation. Authors should discuss and / or mitigate their sentence, also in the dedicated part of the Discussion (lines 1083-1084).

Response:

We have rephrased the sentences in the Discussion (Lines 1193-1199), Conclusion (Lines 1131-1138) and Abstract of the revised manuscript as per your suggestion. Thank you very much.

Minor points

Comment:

Fetal bovine serum is used to culture PBMCs instead of human one: what is the reason why?

Response:

Thank you very much. Your point is very well taken. However, we used fetal bovine serum (FBS) because it is commonly used as a supplement for cell cultures for providing the necessary nutrients for the growth of most cell lines and primary cells. We must confess that we did not apply to the IRB for the use of human serum.

Comment:

“gramzyme” in Fig 8.

Response:

The word “gramzyme” in y axes of Figure 8 was changed to “granzyme. Thank you very much.  

Comment:

Sentence in lines 27-28 is not clear.

Response:

Binding of OX40 ligand (OX40L) at the cysteine-rich domain 2 of extracellular portion of OX40 on T cells has been shown previously to cause downstream signaling in T cells leading to activation of NF-kB transcription factor and gene expression.

Comment:

Sentence in lines 57-58 is not clear.

Response:

We mean immune surveillance by host natural killer cells which is anti-cancer innate immunity at homeostasis

Comment:

Line 86: binds.

Response:

The word “binds” was corrected to “bind” (Line 87 of the revised manuscript). Thank you.

Comment:

Sentence in lines 100-102 is not clear.

Response:

We have rephrased the sentence (Lines 101-102 of the revised manuscript). Thank you.

Comment:

Line 990: co-culturing.

Response:

The word “co-culturing” in Line 990 was changed to co-cultured” in Line 1013-1014 of the revised manuscript. Thank you.

Comment:

In general, higher quality images should be provided.

Responses:

We have changed Figures 1E, 2C, 2D, 6B, 6C, 7A-7C and 10A to improve the quality of the images. All Figures are 700 dpi (as per the journal requirement). Thank you.

Comments on the Quality of English Language

Few minor revisions on English Language are needed (see Minor Points of the revision).

Response:

Thank you very much. We have revised them accordingly.

On behalf of all authors, I would like to thank you for your time spent on reading our manuscript and your leaned comments and suggestions. Our appreciations.

                                                                                 Yours sincerely,

                                                                  Wanpen Chaicumpa, D.V.M. (Hons.), Ph. D.

Reviewer 1

In the proposed manuscript, Mahasongkram and colleagues present the production and characterization of novel human bivalent agonistic fusion antibodies to OX40/CD134. Concerning the production phase, the use of techniques such as phage display and genetic engineering are well explained and robust. However, despite an initial characterization of the antibodies being performed, their functionality is not completely convincing, and some issues are raised in the proposed revision.

In general, the topic is relevant but not novel, as exposed in the Introduction (lines 103-117). The open questions, mainly concerning the effect of the antibodies on Tregs and on signaling cascade, are not fully addressed, and solved by the data.

Major points

Comment:

Fig. 1.

The use of Jurkat cells is not explained, nor detailed in Methods section.

Response:          

We have added the details of Jurkat cells to “Methods” sections 2.2 and 2.4 of the revised manuscript. Thank you very much.

Comment:

Fig 7C.

The selected concentration 1 mg/mL of the fusion antibodies seems suboptimal for C41Fc that may explain its lower activity compared to C2Fc shown in the following figures. What is the reason for choosing the same condition in the two cases, given the different biological outcome?

Response:

We chose this concentration of fusion antibodies because this concentration gave reproductive results and could show the difference between medium control and fusion antibodies. It might be suboptimal as per your comment. We have added your point to the text of the revised manuscript (Lines 890-893 and 1193-1199). Thank you very much.

Comment:

Fig 8.

Even if statistically different, y axis scales of Granzyme B graphs suggest that the observed increases may not have a biological meaning (e.g., from 4 to 7 % of producing T cells). Authors should discuss this point.

Response:

Thank you very much. We have added your point to “Discussion”, i.e., Lines 1193-1199 of the revised manuscript.

Comment:

Differences of basal level values (“medium control”, gray bars) among the staining of the same T cell group should be justified (e.g., from 0.5 to 20% approximately in CD4 T cells group).

Representative gates/histograms should be shown, as in Fig 9A.

Response:

In Figure 8, the basal levels of cytokines in the medium controls were different because different types of cytokines (IL-2, IFN-γ, TNF-α, and granzyme B) can be produced at different levels by different subsets of T cells. We show the percentages of cytokine producing cells in each subset of T cells in in vitro stimulation and the ability of the fusion antibodies in enhancement of cytokine production. The normalization data are presented in fold changes of cytokines compared to respective medium controls.

The representative gates of flow cytometric dot plots are added to new Figure 9 of the revised manuscript. Thus, Figures 9, 10, 11 and 12 of the original manuscript were changed to Figures 10, 11, 12 and 13, respectively, of the revised manuscript.

Thank you very much.

Comment:

Fig 9B.

Bars do not seem to be significantly different, in spite of statistics.

Response:

In this study, we use ANOVA-Kruskal Wallis test. We found that medium control and C2Fc are significantly different (p-value at 0.0306) as shown in the table below.

Live

Dunn's multiple comparisons test

Mean rank diff.

Significant?

Summary

Adjusted p value

Medium vs C2Fc

-11.04

Yes

*

0.0306

Medium vs C41Fc

-3.333

No

ns

> 0.9999

C2Fc vs C41Fc

7.708

No

ns

0.2186

Comment:

Fig 11C.

Data are not completely convincing. Authors state that “tumor antigen primed-T cells exposed to C2Fc/C41Fc fusion antibodies had a trend of increasing ability in tumor cell killing when the effector cell numbers were increased, compared to tumor antigen primed-T cells in medium control (no fusion antibodies)”. This is true for C41Fc in Donor X, while standard deviations of C2Fc are too high to draw any conclusion. Concerning Donor Y, the increasing trend of cancer cell lysis caused by the fusion antibodies is comparable to the one in medium control situation. Authors should discuss and / or mitigate their sentence, also in the dedicated part of the Discussion (lines 1083-1084).

Response:

We have rephrased the sentences in the Discussion (Lines 1193-1199), Conclusion (Lines 1131-1138) and Abstract of the revised manuscript as per your suggestion. Thank you very much.

Minor points

Comment:

Fetal bovine serum is used to culture PBMCs instead of human one: what is the reason why?

Response:

Thank you very much. Your point is very well taken. However, we used fetal bovine serum (FBS) because it is commonly used as a supplement for cell cultures for providing the necessary nutrients for the growth of most cell lines and primary cells. We must confess that we did not apply to the IRB for the use of human serum.

Comment:

“gramzyme” in Fig 8.

Response:

The word “gramzyme” in y axes of Figure 8 was changed to “granzyme. Thank you very much.  

Comment:

Sentence in lines 27-28 is not clear.

Response:

Binding of OX40 ligand (OX40L) at the cysteine-rich domain 2 of extracellular portion of OX40 on T cells has been shown previously to cause downstream signaling in T cells leading to activation of NF-kB transcription factor and gene expression.

Comment:

Sentence in lines 57-58 is not clear.

Response:

We mean immune surveillance by host natural killer cells which is anti-cancer innate immunity at homeostasis

Comment:

Line 86: binds.

Response:

The word “binds” was corrected to “bind” (Line 87 of the revised manuscript). Thank you.

Comment:

Sentence in lines 100-102 is not clear.

Response:

We have rephrased the sentence (Lines 101-102 of the revised manuscript). Thank you.

Comment:

Line 990: co-culturing.

Response:

The word “co-culturing” in Line 990 was changed to co-cultured” in Line 1013-1014 of the revised manuscript. Thank you.

Comment:

In general, higher quality images should be provided.

Responses:

We have changed Figures 1E, 2C, 2D, 6B, 6C, 7A-7C and 10A to improve the quality of the images. All Figures are 700 dpi (as per the journal requirement). Thank you.

Comments on the Quality of English Language

Few minor revisions on English Language are needed (see Minor Points of the revision).

Response:

Thank you very much. We have revised them accordingly.

On behalf of all authors, I would like to thank you for your time spent on reading our manuscript and your leaned comments and suggestions. Our appreciations.

                                                                                 Yours sincerely,

                                                                  Wanpen Chaicumpa, D.V.M. (Hons.), Ph. D.

Reviewer 1

In the proposed manuscript, Mahasongkram and colleagues present the production and characterization of novel human bivalent agonistic fusion antibodies to OX40/CD134. Concerning the production phase, the use of techniques such as phage display and genetic engineering are well explained and robust. However, despite an initial characterization of the antibodies being performed, their functionality is not completely convincing, and some issues are raised in the proposed revision.

In general, the topic is relevant but not novel, as exposed in the Introduction (lines 103-117). The open questions, mainly concerning the effect of the antibodies on Tregs and on signaling cascade, are not fully addressed, and solved by the data.

Major points

Comment:

Fig. 1.

The use of Jurkat cells is not explained, nor detailed in Methods section.

Response:          

We have added the details of Jurkat cells to “Methods” sections 2.2 and 2.4 of the revised manuscript. Thank you very much.

Comment:

Fig 7C.

The selected concentration 1 mg/mL of the fusion antibodies seems suboptimal for C41Fc that may explain its lower activity compared to C2Fc shown in the following figures. What is the reason for choosing the same condition in the two cases, given the different biological outcome?

Response:

We chose this concentration of fusion antibodies because this concentration gave reproductive results and could show the difference between medium control and fusion antibodies. It might be suboptimal as per your comment. We have added your point to the text of the revised manuscript (Lines 890-893 and 1193-1199). Thank you very much.

Comment:

Fig 8.

Even if statistically different, y axis scales of Granzyme B graphs suggest that the observed increases may not have a biological meaning (e.g., from 4 to 7 % of producing T cells). Authors should discuss this point.

Response:

Thank you very much. We have added your point to “Discussion”, i.e., Lines 1193-1199 of the revised manuscript.

Comment:

Differences of basal level values (“medium control”, gray bars) among the staining of the same T cell group should be justified (e.g., from 0.5 to 20% approximately in CD4 T cells group).

Representative gates/histograms should be shown, as in Fig 9A.

Response:

In Figure 8, the basal levels of cytokines in the medium controls were different because different types of cytokines (IL-2, IFN-γ, TNF-α, and granzyme B) can be produced at different levels by different subsets of T cells. We show the percentages of cytokine producing cells in each subset of T cells in in vitro stimulation and the ability of the fusion antibodies in enhancement of cytokine production. The normalization data are presented in fold changes of cytokines compared to respective medium controls.

The representative gates of flow cytometric dot plots are added to new Figure 9 of the revised manuscript. Thus, Figures 9, 10, 11 and 12 of the original manuscript were changed to Figures 10, 11, 12 and 13, respectively, of the revised manuscript.

Thank you very much.

Comment:

Fig 9B.

Bars do not seem to be significantly different, in spite of statistics.

Response:

In this study, we use ANOVA-Kruskal Wallis test. We found that medium control and C2Fc are significantly different (p-value at 0.0306) as shown in the table below.

Live

Dunn's multiple comparisons test

Mean rank diff.

Significant?

Summary

Adjusted p value

Medium vs C2Fc

-11.04

Yes

*

0.0306

Medium vs C41Fc

-3.333

No

ns

> 0.9999

C2Fc vs C41Fc

7.708

No

ns

0.2186

Comment:

Fig 11C.

Data are not completely convincing. Authors state that “tumor antigen primed-T cells exposed to C2Fc/C41Fc fusion antibodies had a trend of increasing ability in tumor cell killing when the effector cell numbers were increased, compared to tumor antigen primed-T cells in medium control (no fusion antibodies)”. This is true for C41Fc in Donor X, while standard deviations of C2Fc are too high to draw any conclusion. Concerning Donor Y, the increasing trend of cancer cell lysis caused by the fusion antibodies is comparable to the one in medium control situation. Authors should discuss and / or mitigate their sentence, also in the dedicated part of the Discussion (lines 1083-1084).

Response:

We have rephrased the sentences in the Discussion (Lines 1193-1199), Conclusion (Lines 1131-1138) and Abstract of the revised manuscript as per your suggestion. Thank you very much.

Minor points

Comment:

Fetal bovine serum is used to culture PBMCs instead of human one: what is the reason why?

Response:

Thank you very much. Your point is very well taken. However, we used fetal bovine serum (FBS) because it is commonly used as a supplement for cell cultures for providing the necessary nutrients for the growth of most cell lines and primary cells. We must confess that we did not apply to the IRB for the use of human serum.

Comment:

“gramzyme” in Fig 8.

Response:

The word “gramzyme” in y axes of Figure 8 was changed to “granzyme. Thank you very much.  

Comment:

Sentence in lines 27-28 is not clear.

Response:

Binding of OX40 ligand (OX40L) at the cysteine-rich domain 2 of extracellular portion of OX40 on T cells has been shown previously to cause downstream signaling in T cells leading to activation of NF-kB transcription factor and gene expression.

Comment:

Sentence in lines 57-58 is not clear.

Response:

We mean immune surveillance by host natural killer cells which is anti-cancer innate immunity at homeostasis

Comment:

Line 86: binds.

Response:

The word “binds” was corrected to “bind” (Line 87 of the revised manuscript). Thank you.

Comment:

Sentence in lines 100-102 is not clear.

Response:

We have rephrased the sentence (Lines 101-102 of the revised manuscript). Thank you.

Comment:

Line 990: co-culturing.

Response:

The word “co-culturing” in Line 990 was changed to co-cultured” in Line 1013-1014 of the revised manuscript. Thank you.

Comment:

In general, higher quality images should be provided.

Responses:

We have changed Figures 1E, 2C, 2D, 6B, 6C, 7A-7C and 10A to improve the quality of the images. All Figures are 700 dpi (as per the journal requirement). Thank you.

Comments on the Quality of English Language

Few minor revisions on English Language are needed (see Minor Points of the revision).

Response:

Thank you very much. We have revised them accordingly.

On behalf of all authors, I would like to thank you for your time spent on reading our manuscript and your leaned comments and suggestions. Our appreciations.

                                                                                 Yours sincerely,

                                                                  Wanpen Chaicumpa, D.V.M. (Hons.), Ph. D.

Reviewer 1

In the proposed manuscript, Mahasongkram and colleagues present the production and characterization of novel human bivalent agonistic fusion antibodies to OX40/CD134. Concerning the production phase, the use of techniques such as phage display and genetic engineering are well explained and robust. However, despite an initial characterization of the antibodies being performed, their functionality is not completely convincing, and some issues are raised in the proposed revision.

In general, the topic is relevant but not novel, as exposed in the Introduction (lines 103-117). The open questions, mainly concerning the effect of the antibodies on Tregs and on signaling cascade, are not fully addressed, and solved by the data.

Major points

Comment:

Fig. 1.

The use of Jurkat cells is not explained, nor detailed in Methods section.

Response:          

We have added the details of Jurkat cells to “Methods” sections 2.2 and 2.4 of the revised manuscript. Thank you very much.

Comment:

Fig 7C.

The selected concentration 1 mg/mL of the fusion antibodies seems suboptimal for C41Fc that may explain its lower activity compared to C2Fc shown in the following figures. What is the reason for choosing the same condition in the two cases, given the different biological outcome?

Response:

We chose this concentration of fusion antibodies because this concentration gave reproductive results and could show the difference between medium control and fusion antibodies. It might be suboptimal as per your comment. We have added your point to the text of the revised manuscript (Lines 890-893 and 1193-1199). Thank you very much.

Comment:

Fig 8.

Even if statistically different, y axis scales of Granzyme B graphs suggest that the observed increases may not have a biological meaning (e.g., from 4 to 7 % of producing T cells). Authors should discuss this point.

Response:

Thank you very much. We have added your point to “Discussion”, i.e., Lines 1193-1199 of the revised manuscript.

Comment:

Differences of basal level values (“medium control”, gray bars) among the staining of the same T cell group should be justified (e.g., from 0.5 to 20% approximately in CD4 T cells group).

Representative gates/histograms should be shown, as in Fig 9A.

Response:

In Figure 8, the basal levels of cytokines in the medium controls were different because different types of cytokines (IL-2, IFN-γ, TNF-α, and granzyme B) can be produced at different levels by different subsets of T cells. We show the percentages of cytokine producing cells in each subset of T cells in in vitro stimulation and the ability of the fusion antibodies in enhancement of cytokine production. The normalization data are presented in fold changes of cytokines compared to respective medium controls.

The representative gates of flow cytometric dot plots are added to new Figure 9 of the revised manuscript. Thus, Figures 9, 10, 11 and 12 of the original manuscript were changed to Figures 10, 11, 12 and 13, respectively, of the revised manuscript.

Thank you very much.

Comment:

Fig 9B.

Bars do not seem to be significantly different, in spite of statistics.

Response:

In this study, we use ANOVA-Kruskal Wallis test. We found that medium control and C2Fc are significantly different (p-value at 0.0306) as shown in the table below.

Live

Dunn's multiple comparisons test

Mean rank diff.

Significant?

Summary

Adjusted p value

Medium vs C2Fc

-11.04

Yes

*

0.0306

Medium vs C41Fc

-3.333

No

ns

> 0.9999

C2Fc vs C41Fc

7.708

No

ns

0.2186

Comment:

Fig 11C.

Data are not completely convincing. Authors state that “tumor antigen primed-T cells exposed to C2Fc/C41Fc fusion antibodies had a trend of increasing ability in tumor cell killing when the effector cell numbers were increased, compared to tumor antigen primed-T cells in medium control (no fusion antibodies)”. This is true for C41Fc in Donor X, while standard deviations of C2Fc are too high to draw any conclusion. Concerning Donor Y, the increasing trend of cancer cell lysis caused by the fusion antibodies is comparable to the one in medium control situation. Authors should discuss and / or mitigate their sentence, also in the dedicated part of the Discussion (lines 1083-1084).

Response:

We have rephrased the sentences in the Discussion (Lines 1193-1199), Conclusion (Lines 1131-1138) and Abstract of the revised manuscript as per your suggestion. Thank you very much.

Minor points

Comment:

Fetal bovine serum is used to culture PBMCs instead of human one: what is the reason why?

Response:

Thank you very much. Your point is very well taken. However, we used fetal bovine serum (FBS) because it is commonly used as a supplement for cell cultures for providing the necessary nutrients for the growth of most cell lines and primary cells. We must confess that we did not apply to the IRB for the use of human serum.

Comment:

“gramzyme” in Fig 8.

Response:

The word “gramzyme” in y axes of Figure 8 was changed to “granzyme. Thank you very much.  

Comment:

Sentence in lines 27-28 is not clear.

Response:

Binding of OX40 ligand (OX40L) at the cysteine-rich domain 2 of extracellular portion of OX40 on T cells has been shown previously to cause downstream signaling in T cells leading to activation of NF-kB transcription factor and gene expression.

Comment:

Sentence in lines 57-58 is not clear.

Response:

We mean immune surveillance by host natural killer cells which is anti-cancer innate immunity at homeostasis

Comment:

Line 86: binds.

Response:

The word “binds” was corrected to “bind” (Line 87 of the revised manuscript). Thank you.

Comment:

Sentence in lines 100-102 is not clear.

Response:

We have rephrased the sentence (Lines 101-102 of the revised manuscript). Thank you.

Comment:

Line 990: co-culturing.

Response:

The word “co-culturing” in Line 990 was changed to co-cultured” in Line 1013-1014 of the revised manuscript. Thank you.

Comment:

In general, higher quality images should be provided.

Responses:

We have changed Figures 1E, 2C, 2D, 6B, 6C, 7A-7C and 10A to improve the quality of the images. All Figures are 700 dpi (as per the journal requirement). Thank you.

Comments on the Quality of English Language

Few minor revisions on English Language are needed (see Minor Points of the revision).

Response:

Thank you very much. We have revised them accordingly.

On behalf of all authors, I would like to thank you for your time spent on reading our manuscript and your leaned comments and suggestions. Our appreciations.

                                                                                 Yours sincerely,

                                                                  Wanpen Chaicumpa, D.V.M. (Hons.), Ph. D.

Reviewer 1

In the proposed manuscript, Mahasongkram and colleagues present the production and characterization of novel human bivalent agonistic fusion antibodies to OX40/CD134. Concerning the production phase, the use of techniques such as phage display and genetic engineering are well explained and robust. However, despite an initial characterization of the antibodies being performed, their functionality is not completely convincing, and some issues are raised in the proposed revision.

In general, the topic is relevant but not novel, as exposed in the Introduction (lines 103-117). The open questions, mainly concerning the effect of the antibodies on Tregs and on signaling cascade, are not fully addressed, and solved by the data.

Major points

Comment:

Fig. 1.

The use of Jurkat cells is not explained, nor detailed in Methods section.

Response:          

We have added the details of Jurkat cells to “Methods” sections 2.2 and 2.4 of the revised manuscript. Thank you very much.

Comment:

Fig 7C.

The selected concentration 1 mg/mL of the fusion antibodies seems suboptimal for C41Fc that may explain its lower activity compared to C2Fc shown in the following figures. What is the reason for choosing the same condition in the two cases, given the different biological outcome?

Response:

We chose this concentration of fusion antibodies because this concentration gave reproductive results and could show the difference between medium control and fusion antibodies. It might be suboptimal as per your comment. We have added your point to the text of the revised manuscript (Lines 890-893 and 1193-1199). Thank you very much.

Comment:

Fig 8.

Even if statistically different, y axis scales of Granzyme B graphs suggest that the observed increases may not have a biological meaning (e.g., from 4 to 7 % of producing T cells). Authors should discuss this point.

Response:

Thank you very much. We have added your point to “Discussion”, i.e., Lines 1193-1199 of the revised manuscript.

Comment:

Differences of basal level values (“medium control”, gray bars) among the staining of the same T cell group should be justified (e.g., from 0.5 to 20% approximately in CD4 T cells group).

Representative gates/histograms should be shown, as in Fig 9A.

Response:

In Figure 8, the basal levels of cytokines in the medium controls were different because different types of cytokines (IL-2, IFN-γ, TNF-α, and granzyme B) can be produced at different levels by different subsets of T cells. We show the percentages of cytokine producing cells in each subset of T cells in in vitro stimulation and the ability of the fusion antibodies in enhancement of cytokine production. The normalization data are presented in fold changes of cytokines compared to respective medium controls.

The representative gates of flow cytometric dot plots are added to new Figure 9 of the revised manuscript. Thus, Figures 9, 10, 11 and 12 of the original manuscript were changed to Figures 10, 11, 12 and 13, respectively, of the revised manuscript.

Thank you very much.

Comment:

Fig 9B.

Bars do not seem to be significantly different, in spite of statistics.

Response:

In this study, we use ANOVA-Kruskal Wallis test. We found that medium control and C2Fc are significantly different (p-value at 0.0306) as shown in the table below.

Live

Dunn's multiple comparisons test

Mean rank diff.

Significant?

Summary

Adjusted p value

Medium vs C2Fc

-11.04

Yes

*

0.0306

Medium vs C41Fc

-3.333

No

ns

> 0.9999

C2Fc vs C41Fc

7.708

No

ns

0.2186

Comment:

Fig 11C.

Data are not completely convincing. Authors state that “tumor antigen primed-T cells exposed to C2Fc/C41Fc fusion antibodies had a trend of increasing ability in tumor cell killing when the effector cell numbers were increased, compared to tumor antigen primed-T cells in medium control (no fusion antibodies)”. This is true for C41Fc in Donor X, while standard deviations of C2Fc are too high to draw any conclusion. Concerning Donor Y, the increasing trend of cancer cell lysis caused by the fusion antibodies is comparable to the one in medium control situation. Authors should discuss and / or mitigate their sentence, also in the dedicated part of the Discussion (lines 1083-1084).

Response:

We have rephrased the sentences in the Discussion (Lines 1193-1199), Conclusion (Lines 1131-1138) and Abstract of the revised manuscript as per your suggestion. Thank you very much.

Minor points

Comment:

Fetal bovine serum is used to culture PBMCs instead of human one: what is the reason why?

Response:

Thank you very much. Your point is very well taken. However, we used fetal bovine serum (FBS) because it is commonly used as a supplement for cell cultures for providing the necessary nutrients for the growth of most cell lines and primary cells. We must confess that we did not apply to the IRB for the use of human serum.

Comment:

“gramzyme” in Fig 8.

Response:

The word “gramzyme” in y axes of Figure 8 was changed to “granzyme. Thank you very much.  

Comment:

Sentence in lines 27-28 is not clear.

Response:

Binding of OX40 ligand (OX40L) at the cysteine-rich domain 2 of extracellular portion of OX40 on T cells has been shown previously to cause downstream signaling in T cells leading to activation of NF-kB transcription factor and gene expression.

Comment:

Sentence in lines 57-58 is not clear.

Response:

We mean immune surveillance by host natural killer cells which is anti-cancer innate immunity at homeostasis

Comment:

Line 86: binds.

Response:

The word “binds” was corrected to “bind” (Line 87 of the revised manuscript). Thank you.

Comment:

Sentence in lines 100-102 is not clear.

Response:

We have rephrased the sentence (Lines 101-102 of the revised manuscript). Thank you.

Comment:

Line 990: co-culturing.

Response:

The word “co-culturing” in Line 990 was changed to co-cultured” in Line 1013-1014 of the revised manuscript. Thank you.

Comment:

In general, higher quality images should be provided.

Responses:

We have changed Figures 1E, 2C, 2D, 6B, 6C, 7A-7C and 10A to improve the quality of the images. All Figures are 700 dpi (as per the journal requirement). Thank you.

Comments on the Quality of English Language

Few minor revisions on English Language are needed (see Minor Points of the revision).

Response:

Thank you very much. We have revised them accordingly.

On behalf of all authors, I would like to thank you for your time spent on reading our manuscript and your leaned comments and suggestions. Our appreciations.

                                                                                 Yours sincerely,

                                                                  Wanpen Chaicumpa, D.V.M. (Hons.), Ph. D.

Reviewer 1

In the proposed manuscript, Mahasongkram and colleagues present the production and characterization of novel human bivalent agonistic fusion antibodies to OX40/CD134. Concerning the production phase, the use of techniques such as phage display and genetic engineering are well explained and robust. However, despite an initial characterization of the antibodies being performed, their functionality is not completely convincing, and some issues are raised in the proposed revision.

In general, the topic is relevant but not novel, as exposed in the Introduction (lines 103-117). The open questions, mainly concerning the effect of the antibodies on Tregs and on signaling cascade, are not fully addressed, and solved by the data.

Major points

Comment:

Fig. 1.

The use of Jurkat cells is not explained, nor detailed in Methods section.

Response:          

We have added the details of Jurkat cells to “Methods” sections 2.2 and 2.4 of the revised manuscript. Thank you very much.

Comment:

Fig 7C.

The selected concentration 1 mg/mL of the fusion antibodies seems suboptimal for C41Fc that may explain its lower activity compared to C2Fc shown in the following figures. What is the reason for choosing the same condition in the two cases, given the different biological outcome?

Response:

We chose this concentration of fusion antibodies because this concentration gave reproductive results and could show the difference between medium control and fusion antibodies. It might be suboptimal as per your comment. We have added your point to the text of the revised manuscript (Lines 890-893 and 1193-1199). Thank you very much.

Comment:

Fig 8.

Even if statistically different, y axis scales of Granzyme B graphs suggest that the observed increases may not have a biological meaning (e.g., from 4 to 7 % of producing T cells). Authors should discuss this point.

Response:

Thank you very much. We have added your point to “Discussion”, i.e., Lines 1193-1199 of the revised manuscript.

Comment:

Differences of basal level values (“medium control”, gray bars) among the staining of the same T cell group should be justified (e.g., from 0.5 to 20% approximately in CD4 T cells group).

Representative gates/histograms should be shown, as in Fig 9A.

Response:

In Figure 8, the basal levels of cytokines in the medium controls were different because different types of cytokines (IL-2, IFN-γ, TNF-α, and granzyme B) can be produced at different levels by different subsets of T cells. We show the percentages of cytokine producing cells in each subset of T cells in in vitro stimulation and the ability of the fusion antibodies in enhancement of cytokine production. The normalization data are presented in fold changes of cytokines compared to respective medium controls.

The representative gates of flow cytometric dot plots are added to new Figure 9 of the revised manuscript. Thus, Figures 9, 10, 11 and 12 of the original manuscript were changed to Figures 10, 11, 12 and 13, respectively, of the revised manuscript.

Thank you very much.

Comment:

Fig 9B.

Bars do not seem to be significantly different, in spite of statistics.

Response:

In this study, we use ANOVA-Kruskal Wallis test. We found that medium control and C2Fc are significantly different (p-value at 0.0306) as shown in the table below.

Live

Dunn's multiple comparisons test

Mean rank diff.

Significant?

Summary

Adjusted p value

Medium vs C2Fc

-11.04

Yes

*

0.0306

Medium vs C41Fc

-3.333

No

ns

> 0.9999

C2Fc vs C41Fc

7.708

No

ns

0.2186

Comment:

Fig 11C.

Data are not completely convincing. Authors state that “tumor antigen primed-T cells exposed to C2Fc/C41Fc fusion antibodies had a trend of increasing ability in tumor cell killing when the effector cell numbers were increased, compared to tumor antigen primed-T cells in medium control (no fusion antibodies)”. This is true for C41Fc in Donor X, while standard deviations of C2Fc are too high to draw any conclusion. Concerning Donor Y, the increasing trend of cancer cell lysis caused by the fusion antibodies is comparable to the one in medium control situation. Authors should discuss and / or mitigate their sentence, also in the dedicated part of the Discussion (lines 1083-1084).

Response:

We have rephrased the sentences in the Discussion (Lines 1193-1199), Conclusion (Lines 1131-1138) and Abstract of the revised manuscript as per your suggestion. Thank you very much.

Minor points

Comment:

Fetal bovine serum is used to culture PBMCs instead of human one: what is the reason why?

Response:

Thank you very much. Your point is very well taken. However, we used fetal bovine serum (FBS) because it is commonly used as a supplement for cell cultures for providing the necessary nutrients for the growth of most cell lines and primary cells. We must confess that we did not apply to the IRB for the use of human serum.

Comment:

“gramzyme” in Fig 8.

Response:

The word “gramzyme” in y axes of Figure 8 was changed to “granzyme. Thank you very much.  

Comment:

Sentence in lines 27-28 is not clear.

Response:

Binding of OX40 ligand (OX40L) at the cysteine-rich domain 2 of extracellular portion of OX40 on T cells has been shown previously to cause downstream signaling in T cells leading to activation of NF-kB transcription factor and gene expression.

Comment:

Sentence in lines 57-58 is not clear.

Response:

We mean immune surveillance by host natural killer cells which is anti-cancer innate immunity at homeostasis

Comment:

Line 86: binds.

Response:

The word “binds” was corrected to “bind” (Line 87 of the revised manuscript). Thank you.

Comment:

Sentence in lines 100-102 is not clear.

Response:

We have rephrased the sentence (Lines 101-102 of the revised manuscript). Thank you.

Comment:

Line 990: co-culturing.

Response:

The word “co-culturing” in Line 990 was changed to co-cultured” in Line 1013-1014 of the revised manuscript. Thank you.

Comment:

In general, higher quality images should be provided.

Responses:

We have changed Figures 1E, 2C, 2D, 6B, 6C, 7A-7C and 10A to improve the quality of the images. All Figures are 700 dpi (as per the journal requirement). Thank you.

Comments on the Quality of English Language

Few minor revisions on English Language are needed (see Minor Points of the revision).

Response:

Thank you very much. We have revised them accordingly.

On behalf of all authors, I would like to thank you for your time spent on reading our manuscript and your leaned comments and suggestions. Our appreciations.

                                                                                 Yours sincerely,

                                                                  Wanpen Chaicumpa, D.V.M. (Hons.), Ph. D.

Reviewer 1

In the proposed manuscript, Mahasongkram and colleagues present the production and characterization of novel human bivalent agonistic fusion antibodies to OX40/CD134. Concerning the production phase, the use of techniques such as phage display and genetic engineering are well explained and robust. However, despite an initial characterization of the antibodies being performed, their functionality is not completely convincing, and some issues are raised in the proposed revision.

In general, the topic is relevant but not novel, as exposed in the Introduction (lines 103-117). The open questions, mainly concerning the effect of the antibodies on Tregs and on signaling cascade, are not fully addressed, and solved by the data.

Major points

Comment:

Fig. 1.

The use of Jurkat cells is not explained, nor detailed in Methods section.

Response:          

We have added the details of Jurkat cells to “Methods” sections 2.2 and 2.4 of the revised manuscript. Thank you very much.

Comment:

Fig 7C.

The selected concentration 1 mg/mL of the fusion antibodies seems suboptimal for C41Fc that may explain its lower activity compared to C2Fc shown in the following figures. What is the reason for choosing the same condition in the two cases, given the different biological outcome?

Response:

We chose this concentration of fusion antibodies because this concentration gave reproductive results and could show the difference between medium control and fusion antibodies. It might be suboptimal as per your comment. We have added your point to the text of the revised manuscript (Lines 890-893 and 1193-1199). Thank you very much.

Comment:

Fig 8.

Even if statistically different, y axis scales of Granzyme B graphs suggest that the observed increases may not have a biological meaning (e.g., from 4 to 7 % of producing T cells). Authors should discuss this point.

Response:

Thank you very much. We have added your point to “Discussion”, i.e., Lines 1193-1199 of the revised manuscript.

Comment:

Differences of basal level values (“medium control”, gray bars) among the staining of the same T cell group should be justified (e.g., from 0.5 to 20% approximately in CD4 T cells group).

Representative gates/histograms should be shown, as in Fig 9A.

Response:

In Figure 8, the basal levels of cytokines in the medium controls were different because different types of cytokines (IL-2, IFN-γ, TNF-α, and granzyme B) can be produced at different levels by different subsets of T cells. We show the percentages of cytokine producing cells in each subset of T cells in in vitro stimulation and the ability of the fusion antibodies in enhancement of cytokine production. The normalization data are presented in fold changes of cytokines compared to respective medium controls.

The representative gates of flow cytometric dot plots are added to new Figure 9 of the revised manuscript. Thus, Figures 9, 10, 11 and 12 of the original manuscript were changed to Figures 10, 11, 12 and 13, respectively, of the revised manuscript.

Thank you very much.

Comment:

Fig 9B.

Bars do not seem to be significantly different, in spite of statistics.

Response:

In this study, we use ANOVA-Kruskal Wallis test. We found that medium control and C2Fc are significantly different (p-value at 0.0306) as shown in the table below.

Live

Dunn's multiple comparisons test

Mean rank diff.

Significant?

Summary

Adjusted p value

Medium vs C2Fc

-11.04

Yes

*

0.0306

Medium vs C41Fc

-3.333

No

ns

> 0.9999

C2Fc vs C41Fc

7.708

No

ns

0.2186

Comment:

Fig 11C.

Data are not completely convincing. Authors state that “tumor antigen primed-T cells exposed to C2Fc/C41Fc fusion antibodies had a trend of increasing ability in tumor cell killing when the effector cell numbers were increased, compared to tumor antigen primed-T cells in medium control (no fusion antibodies)”. This is true for C41Fc in Donor X, while standard deviations of C2Fc are too high to draw any conclusion. Concerning Donor Y, the increasing trend of cancer cell lysis caused by the fusion antibodies is comparable to the one in medium control situation. Authors should discuss and / or mitigate their sentence, also in the dedicated part of the Discussion (lines 1083-1084).

Response:

We have rephrased the sentences in the Discussion (Lines 1193-1199), Conclusion (Lines 1131-1138) and Abstract of the revised manuscript as per your suggestion. Thank you very much.

Minor points

Comment:

Fetal bovine serum is used to culture PBMCs instead of human one: what is the reason why?

Response:

Thank you very much. Your point is very well taken. However, we used fetal bovine serum (FBS) because it is commonly used as a supplement for cell cultures for providing the necessary nutrients for the growth of most cell lines and primary cells. We must confess that we did not apply to the IRB for the use of human serum.

Comment:

“gramzyme” in Fig 8.

Response:

The word “gramzyme” in y axes of Figure 8 was changed to “granzyme. Thank you very much.  

Comment:

Sentence in lines 27-28 is not clear.

Response:

Binding of OX40 ligand (OX40L) at the cysteine-rich domain 2 of extracellular portion of OX40 on T cells has been shown previously to cause downstream signaling in T cells leading to activation of NF-kB transcription factor and gene expression.

Comment:

Sentence in lines 57-58 is not clear.

Response:

We mean immune surveillance by host natural killer cells which is anti-cancer innate immunity at homeostasis

Comment:

Line 86: binds.

Response:

The word “binds” was corrected to “bind” (Line 87 of the revised manuscript). Thank you.

Comment:

Sentence in lines 100-102 is not clear.

Response:

We have rephrased the sentence (Lines 101-102 of the revised manuscript). Thank you.

Comment:

Line 990: co-culturing.

Response:

The word “co-culturing” in Line 990 was changed to co-cultured” in Line 1013-1014 of the revised manuscript. Thank you.

Comment:

In general, higher quality images should be provided.

Responses:

We have changed Figures 1E, 2C, 2D, 6B, 6C, 7A-7C and 10A to improve the quality of the images. All Figures are 700 dpi (as per the journal requirement). Thank you.

Comments on the Quality of English Language

Few minor revisions on English Language are needed (see Minor Points of the revision).

Response:

Thank you very much. We have revised them accordingly.

On behalf of all authors, I would like to thank you for your time spent on reading our manuscript and your leaned comments and suggestions. Our appreciations.

                                                                                 Yours sincerely,

                                                                  Wanpen Chaicumpa, D.V.M. (Hons.), Ph. D.

Reviewer 2 Report

Comments and Suggestions for Authors

Kodchakorn Mahasongkram and colleagues in their study focused on the development of novel agonistic antibodies targeting the OX40/CD134 ectodomain (EcOX40), specifically fully human bivalent single-chain variable fragments (HuscFvs) linked to IgG Fc (bivalent HuscFvs-Fcγ fusion antibodies). Utilizing phage display technology and genetic engineering, these fusion antibodies were designed to bind to the cysteine-rich domain-2 of EcOX40, known to be involved in OX40-OX40L signaling for NF-kB activation. The fusion antibodies demonstrated the ability to induce proliferation, enhance survival, and promote cytokine/granzyme B production in activated human T cells. Furthermore, they exhibited an increased cytotoxicity of tumor-reactive T cells against ovarian cancer cells.

Their findings suggest that the novel OX40 agonistic fusion antibodies hold promise and merit further systematic evaluation for their safety and potential as a non-immunogenic cancer immunotherapeutic agent.

There is substantial experimental work carried out with Figures demonstrating the findings, Interpretation of the results and sufficient Discussion of the limitations and further work to be carried out.

Concerning the computational work that I can better evaluate, the molecular modelling approach is adequate and credible.

To enhance the study's completeness, it is recommended that the authors incorporate a reference to IMGT (the international ImMunoGeneTics information system), accessible at https://academic.oup.com/nar/article/50/D1/D1262/6455007?login=false. Additionally, the authors are encouraged to leverage IMGT/mAb-DB and provide a citation (https://www.frontiersin.org/articles/10.3389/fimmu.2023.1129323/full). This database facilitates the graphical standardization of the mechanisms of action for various monoclonal antibodies, a couple of them mentioned in their Introduction.

These additions would contribute to a more comprehensive and well-referenced exploration of the molecular aspects of the study, aligning it with established resources and enhancing its credibility.

Author Response

Response to Reviewer 2’s Comments

Kodchakorn Mahasongkram and colleagues in their study focused on the development of novel agonistic antibodies targeting the OX40/CD134 ectodomain (EcOX40), specifically fully human bivalent single-chain variable fragments (HuscFvs) linked to IgG Fc (bivalent HuscFvs-Fcγ fusion antibodies). Utilizing phage display technology and genetic engineering, these fusion antibodies were designed to bind to the cysteine-rich domain-2 of EcOX40, known to be involved in OX40-OX40L signaling for NF-kB activation. The fusion antibodies demonstrated the ability to induce proliferation, enhance survival, and promote cytokine/granzyme B production in activated human T cells. Furthermore, they exhibited an increased cytotoxicity of tumor-reactive T cells against ovarian cancer cells.

Their findings suggest that the novel OX40 agonistic fusion antibodies hold promise and merit further systematic evaluation for their safety and potential as a non-immunogenic cancer immunotherapeutic agent.

There is substantial experimental work carried out with Figures demonstrating the findings, Interpretation of the results and sufficient Discussion of the limitations and further work to be carried out.

Concerning the computational work that I can better evaluate, the molecular modelling approach is adequate and credible.

To enhance the study's completeness, it is recommended that the authors incorporate a reference to IMGT (the international ImMunoGeneTics information system), accessible at https://academic.oup.com/nar/article/50/D1/D1262/6455007?login=false. Additionally, the authors are encouraged to leverage IMGT/mAb-DB and provide a citation (https://www.frontiersin.org/articles/10.3389/fimmu.2023.1129323/full). This database facilitates the graphical standardization of the mechanisms of action for various monoclonal antibodies, a couple of them mentioned in their Introduction.

These additions would contribute to a more comprehensive and well-referenced exploration of the molecular aspects of the study, aligning it with established resources and enhancing its credibility.

Response:

Thank you very much for your kindest comments. Please receive our appreciations.

Your advice on the computational work is well taken with thanks. We have added https://academic.oup.com/nar/article/50/D1/D1262/6455007?login=false. Line 350-351 and Reference 47 of the revised manuscript. Thank you very much.

On behalf of all authors, I would like to thank you for your time spent on reading our manuscript and your leaned comments and suggestions. Our appreciations.

                                                                                 Yours sincerely,

                                                             Wanpen Chaicumpa, D.V.M. (Hons.), Ph. D.

Round 2

Reviewer 1 Report

Comments and Suggestions for Authors

Authors answered to the questions raised in the review.